# Chromatin-bound CRM1 recruits SET-Nup214 and NPM1c onto *HOX* clusters causing aberrant *HOX* expression in leukemia cells

Masahiro Oka[1,2]*, Sonoko Mura[3], Mayumi Otani[1], Yoichi Miyamoto[1], Jumpei Nogami[4], Kazumitsu Maehara[4], Akihito Harada[4], Taro Tachibana[5], Yoshihiro Yoneda[2,6], Yasuyuki Ohkawa[4]

[1]Laboratory of Nuclear Transport Dynamics, National Institutes of Biomedical Innovation, Health and Nutrition (NIBIOHN), Osaka, Japan; [2]Laboratory of Biomedical Innovation, Graduate School of Pharmaceutical Sciences, Osaka University, Osaka, Japan; [3]Biomolecular Dynamics Group, Graduate School of Frontier Biosciences, Osaka University, Osaka, Japan; [4]Department of Advanced Medical Initiatives, Faculty of Medicine, Kyushu University, Fukuoka, Japan; [5]Department of Bioengineering, Graduate School of Engineering, Osaka City University, Osaka, Japan; [6]National Institutes of Biomedical Innovation, Health and Nutrition (NIBIOHN), Osaka, Japan

**\*For correspondence:**
moka@nibiohn.go.jp

**Competing interests:** The authors declare that no competing interests exist.

**Abstract** We previously demonstrated that CRM1, a major nuclear export factor, accumulates at *Hox* cluster regions to recruit nucleoporin-fusion protein Nup98HoxA9, resulting in robust activation of *Hox* genes (Oka et al., 2016). However, whether this phenomenon is general to other leukemogenic proteins remains unknown. Here, we show that two other leukemogenic proteins, nucleoporin-fusion SET-Nup214 and the NPM1 mutant, NPM1c, which contains a nuclear export signal (NES) at its C-terminus and is one of the most frequent mutations in acute myeloid leukemia, are recruited to the *HOX* cluster region via chromatin-bound CRM1, leading to *HOX* gene activation in human leukemia cells. Furthermore, we demonstrate that this mechanism is highly sensitive to a CRM1 inhibitor in leukemia cell line. Together, these findings indicate that CRM1 acts as a key molecule that connects leukemogenic proteins to aberrant *HOX* gene regulation either via nucleoporin-CRM1 interaction (for SET-Nup214) or NES-CRM1 interaction (for NPM1c).

## Introduction

Nucleoporin genes, *Nup98* and *Nup214*, are often rearranged in leukemia and generate fusion genes that cause the development of the disease (*Xu and Powers, 2009*). As for Nup98 partner genes, more than 30 different genes with various functions, such as transcription factor, RNA helicase, DNA topoisomerase, and histone methyltransferase, are known (*Gough et al., 2011*). Nup98-fusions are found most frequently in acute myeloid leukemia (AML), while Nup214-fusions are detected in T-cell acute lymphoblastic leukemia (T-ALL) as well as in AML (*Zhou and Yang, 2014*; *Mendes and Fahrenkrog, 2019*).

So far, the precise pathogenic mechanism of Nup98- or Nup214-fusion–mediated leukemia still remains to be elucidated. However, one of the common features of both Nup98 and Nup214-fusion–expressing leukemia is aberrant activation of *HOX* genes (*Gough et al., 2011*; *Van Vlierberghe et al., 2008*; *Wang et al., 2007*; *Hollink et al., 2011*). *HOX* genes encode

DNA-binding proteins that specify cell fate along the head-tail axis (*Krumlauf, 1994*; *Mallo et al., 2010*). It is also well known that aberrant regulation of *HOX* genes plays a crucial role in leukemogenesis (*Argiropoulos and Humphries, 2007*; *Alharbi et al., 2013*). Previously, we demonstrated that Nup98HoxA9 significantly accumulates on the *Hox* cluster regions when expressed in mouse embryonic stem (ES) cells to aberrantly activate broad *Hox* cluster genes (*Oka et al., 2016*). Subsequently, it was shown that Nup98HoxA9 actually binds to the *Hox* cluster region in immortalized hematopoietic cells (*Shima et al., 2017*; *Xu et al., 2016*). These results suggest that selective targeting of Nup98HoxA9 to the *Hox* cluster region is a fundamental mechanism to induce aberrant *HOX* gene expression and to perturb hematopoiesis.

Furthermore, we previously reported (*Oka et al., 2016*) that CRM1 (chromosomal region maintenance 1, also called exportin-1 or XPO1), a major nuclear export factor (*Fornerod et al., 1997*; *Fukuda et al., 1997*; *Ossareh-Nazari et al., 1997*; *Stade et al., 1997*) which was originally identified in a fission yeast (*Adachi and Yanagida, 1989*), exists in specific chromatin regions including *Hox* gene clusters in the nucleus that are highly correlated with genome wide–binding sites of Nup98HoxA9, suggesting that CRM1 is the molecule that recruits Nup98-fusion protein onto the *HOX* cluster region.

In addition, it was also demonstrated that the Nup214-fusion protein binds to CRM1 (*Saito et al., 2004*; *Saito et al., 2016*; *Port et al., 2016*). Furthermore, it was shown that leukemia cells expressing Nup214-fusion are known to be associated with a high *HOX* gene expression profile (*Van Vlierberghe et al., 2008*). These results collectively suggest that there may exist a common pathogenic mechanism of the major nucleoporin fusions – Nup98- and Nup214-fusions, in leukemia; that is, these fusions may be recruited to the *HOX* cluster regions via chromatin-bound CRM1 to activate *HOX* genes.

Another leukemogenic protein which might be related to chromatin-bound CRM1 is nucleophosmin 1 (NPM1), a multifunctional nucleolar protein that is frequently overexpressed or mutated in human cancers (*Grisendi et al., 2006*). It has been shown that a mutant form of NPM1 is one of the most frequently acquired molecular abnormality, found in approximately one-third of patients with AML (*Falini et al., 2005*; *Verhaak et al., 2005*). This NPM1 mutant (called NPM1c) contains a novel nuclear export signal (NES) at its C-terminus, which is generated by insertion or deletion of nucleotides at C-terminus that causes a frameshift of the downstream open reading frame. Indeed, NPM1c is exported from the nucleus to the cytoplasm in a CRM1-dependent manner (*Falini et al., 2006*; *Bolli et al., 2007*). Interestingly, *HOX* gene activation has been shown in a patient with AML and in a cell line expressing NPM1c (*Alcalay et al., 2005*; *Mullighan et al., 2007*), and that *HOX* gene expression is critical for the growth of NPM1c-expressing cells (*Dovey et al., 2017*; *Brunetti et al., 2018*). Furthermore, the *HOX* gene expression is directly dependent on NPM1c (*Brunetti et al., 2018*).

NPM1 is a multifunctional protein involved in many cellular processes, including the regulation of ARF tumor suppressor (*Itahana et al., 2003*; *Korgaonkar et al., 2005*), histone chaperoning (*Okuwaki et al., 2001*), ribosome biogenesis (*Savkur and Olson, 1998*; *Maggi et al., 2008*; *Murano et al., 2008*), centrosome duplication (*Okuda et al., 2000*), transcriptional regulation, and DNA repair (reviewed in *Grisendi et al., 2006*; *Lindström, 2011*). Certainly, various defects that could potentially cause the disease, which are mainly attributed to the cytoplasmic translocation of NPM1-binding proteins, have been observed in NPM1c-expressing cells (*Colombo et al., 2006*; *den Besten et al., 2005*; *Bonetti et al., 2008*; *Noguera et al., 2013*; *Ando et al., 2017*; *Zou et al., 2017*; *Gu et al., 2018*). However, the relevance of these defects to the pathogenesis still remains to be established.

In this study, we demonstrated that both the SET-Nup214 fusion and NPM1c bind to *HOX* cluster regions to activate *HOX* genes in a CRM1-dependent manner. These results demonstrate that chromatin-bound CRM1 functions as a platform to recruit a broad range of leukemic proteins, either via nucleoporin-CRM1 or NES-CRM1 interaction, resulting in abnormal *HOX* gene expression that causes leukemia.

## Results and discussion

### SET-Nup214 fusion binds to *HOXA* and *HOXB* clusters together with CRM1 in a leukemia cell line

Although we previously reported that Nup98HoxA9 accumulates onto four *Hox* cluster regions (*HoxA*, *HoxB*, *HoxC*, and *HoxD*) via chromatin-bound CRM1 in mouse ES cells (*Oka et al., 2016*), it still remains elusive whether other leukemogenic proteins also accumulate onto *HOX* cluster regions in a CRM1-dependent manner in human leukemia cells. Therefore, in this study, we first focused on SET-Nup214, a fusion between histone chaperone SET and Nup214 (*von Lindern et al., 1992a*), since it was previously shown to bind the promoter region of *HOXA9* and *HOXA10* (*Van Vlierberghe et al., 2008*) and to form nuclear dots that colocalize with CRM1 (*Saito et al., 2004*; *Saito et al., 2016*; *Port et al., 2016*), as observed in Nup98HoxA9 (*Oka et al., 2010*; *Oka et al., 2016*; *Takeda et al., 2010*). However, so far, the binding was only examined by ChIP-qPCR and restricted to the promoter regions of several *HOX* genes. In addition, the genome-wide binding pattern of SET-Nup214 is still unknown.

Therefore, we performed ChIP-sequencing (ChIP-seq) analysis to examine the genome-wide binding profile of CRM1 and SET-Nup214 in the T-ALL cell line, LOUCY. As controls, we used other leukemia cell lines, HL60 and K562. As shown in *Figure 1A–C*, we found that CRM1 selectively binds to *HOXA* and *HOXB* cluster regions, especially to *HOXB* region, in LOUCY cells, while no accumulation of CRM1 was observed in K562 cells or in HL60 cells (*Figure 3—figure supplement 5*). The specific binding of CRM1 onto *HOXB* in LOUCY cells was obvious even at a genome-wide scale (*Figure 1A*) or in a whole chromosome (*Figure 1B–C*; Chr.17 for *HOXB* and Chr.7 for *HOXA*). The binding profile of CRM1 was confirmed by two different anti-CRM1 antibodies from different companies [rabbit monoclonal antibody #46249 from Cell Signaling Technology (CST) and rabbit polyclonal antibody # A300-469A from Bethyl Laboratories] (*Figure 1—figure supplement 1*). Furthermore, ChIP-seq with anti-Nup214 antibody, which reacts with SET-Nup214 fusion (*Figure 3A*, *Figure 3—figure supplement 5A*), showed robust signals on the *HOXA* and *HOXB* cluster regions with a highly similar pattern to that of CRM1, in LOUCY cells (*Figure 2A–C*). In addition, we found that CRM1 signal was enriched at the Nup214 peaks (*Figure 2D*, right panel), while Nup214 signal was not significantly so at the CRM1 peaks (*Figure 2D*, left panel). Notably, we also found that CRM1/SET-Nup214 accumulated in other regions, including *CDKN2C* (Chr.1) and *BMI1* (Chr.10) in LOUCY cells (*Figure 1—figure supplement 2*, *Figure 2—figure supplement 1*). BMI1 is known to be overexpressed in a variety of cancers, including hematopoietic malignancies, and is associated with poor prognosis (*Valk-Lingbeek et al., 2004*). These results indicate that SET-Nup214 is preferentially targeted to a subset of CRM1 binding sites, including *HOX* cluster regions, in LOUCY cells and suggest that SET-Nup214 functions in a similar manner to Nup98HoxA9 in the activation of *HOX* genes.

### CRM1 is required for the recruitment of SET-Nup214 on *HOX* clusters and *HOX* gene activation

We next examined the effect of CRM1 inhibitor, KPT-330, a selective inhibitor of nuclear export (SINE), on the accumulation of SET-Nup214 on the *HOX* clusters. SINEs not only inhibit the binding between CRM1 and NES substrate (*Lapalombella et al., 2012*; *Etchin et al., 2013*) but also cause degradation of CRM1 (*Turner et al., 2013*; *Zhang et al., 2013*; *Mendonca et al., 2014*; *Tai et al., 2014*) (*Figure 3A*). Unexpectedly, we found that SET-Nup214, but not endogenous Nup214, in LOUCY cells was also degraded by treatment with KPT-330 (*Figure 3A*). We detected comparable amounts of CRM1 and Nup214 proteins in HL60 cell lysates as compared with those in LOUCY cells (*Figure 3A*), all of which were prepared by boiling in a sample buffer. However, we noticed that both Nup214 and CRM1 were hardly detected in HL60 cell lysates when they were extracted using RIPA buffer (*Figure 3—figure supplement 5A*). Thus, this unexpected behavior of CRM1 in HL60 cells may explain the low CRM1 ChIP signals (*Figure 3—figure supplement 5B–D*). Next, we performed immunostaining with anti-Nup214 and anti-CRM1. Consistent with a previous report (*Port et al., 2016*), we observed distinct nuclear dots of SET-Nup214 in LOUCY cells that colocalize with CRM1 (*Figure 3—figure supplement 1B*). These dots were not observed in K562 and HL60 cells (*Figure 3—figure supplement 1A*). Furthermore, we found that the SET-Nup214 nuclear dots in LOUCY cells disappeared in the presence of KPT-330 (*Figure 3—figure supplement 1C*),

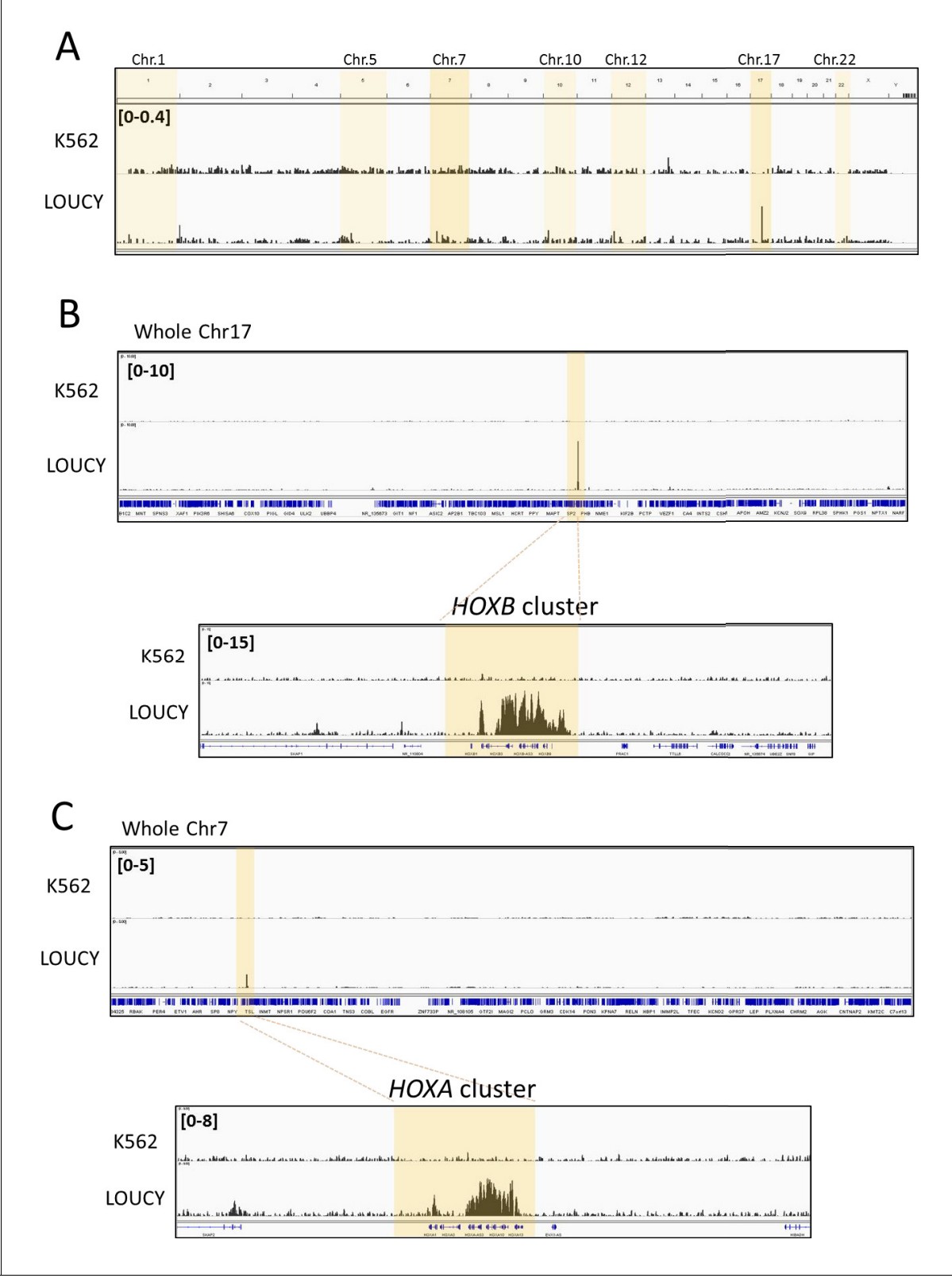

**Figure 1.** CRM1 accumulates on specific genome regions, including *HOXA* and *HOXB* cluster, in human Leukemia cells. Binding profiles of CRM1 in K562 and LOUCY cell lines (**A**: whole genome; **B**: whole chromosome 17 and *HOXB* cluster, **C**: whole chromosome seven and *HOXA* cluster). ChIP-seq was performed using anti-CRM1 (CST, D6V7N, #46249) antibody. In (**A**), the chromosomes that contain strong CRM1 binding sites are shadowed (dark

*Figure 1 continued on next page*

*Figure 1 continued*

shadow; Chr.7 (*HOXA*) and Chr.17 (*HOXB*), light shadow; Chr.1 (*CDKN2C*), Chr.5 (*LOC648987*), Chr.10 (*NR_038921, COMMD3-BMI1*), Chr.12 (*CLEC2B-KLRF2*), Chr.22 (*PRR34*), see *Figure 1—figure supplement 2* for details).

The online version of this article includes the following figure supplement(s) for figure 1:

**Figure supplement 1.** Two different anti-CRM1 antibodies show similar binding profiles of CRM1.

**Figure supplement 2.** CRM1 accumulates on specific genome regions.

consistent with a previous study using leptomycin B (LMB) (*Port et al., 2016*), another well-known CRM1 inhibitor (*Kudo et al., 1998*).

We next performed ChIP-seq analysis to examine the effect of KPT-330 on the accumulation of CRM1 and SET-Nup214 on the *HOX* clusters. We found that the robust peaks of CRM1 and Nup214 almost completely disappeared in LOUCY cells by treatment with 1000 nM of KPT-330 for 24 hr (*Figure 2A–C*, *Figure 2—figure supplement 1*). Subsequently, we monitored gene expression in these leukemia cell lines to study whether the expression of *HOX* genes is affected by treatment with KPT-330. Robust *HOX* gene activation was observed in both the K562 and LOUCY cells but not in HL60 cells (*Figure 3B*). Furthermore, the expression of *HOX* genes in the LOUCY cells was more sensitive to the KPT-330 treatment than that in K562 cells (*Figure 3C*).

We also examined other AML cell lines either expressing SET-Nup214 (MEGAL), or not (HEL) (*Figure 3—figure supplement 2*). CRM1 was similarly sensitive to KPT-330 treatment in both cell lines (*Figure 3—figure supplement 2A–B*). SET-Nup214 expressed in MEGAL also showed sensitivity toward KPT-330 treatment, but the effect was relatively smaller as compared with that in LOUCY cells. Gene expression analysis shows that *HOXA9* genes were highly expressed in MEGAL cells, while *HOXB* genes were highly expressed in HEL cells (*Figure 3—figure supplement 2C*). As expected, *HOXA* genes in MEGAL cells showed sensitivity against KPT-330 treatment, even though the effect was mild as compared with that in LOUCY cells (*Figure 3—figure supplement 2D*). Furthermore, ChIP-qPCR revealed that both CRM1 and SET-Nup214 preferentially bound to *HOXA9* region in MEGAL cells (*Figure 3—figure supplement 2E–F*). Therefore, the binding of CRM1/SET-Nup214 showed a correlation with gene activation in MEGAL cells, as also observed in LOUCY cells. The difference between LOUCY and MEGAL cells in the degree and region of CRM1/SET-Nup214 binding and gene activation is likely related to the fact that MEGAL cells show lower expression of SET-Nup214 than LOUCY cells (comparing the ratio of SET-Nup214/Nup214 in *Figure 3—figure supplement 5A* [LOUCY] and *Figure 3—figure supplement 2A* [MEGAL]). However, the reason for the lower sensitivity of SET-Nup214 in MEGAL cells to KPT330 remains unknown. We also detected some KPT-330-sensitive CRM1 binding signals at *HOX* regions in HEL cells. However, its biological significance remains to be determined.

Collectively, these results demonstrated that both in the LOUCY and MEGAL cells the recruitment of SET-Nup214 fusion to the *HOX* cluster regions is required for the efficient activation of, at least, some *HOX* genes.

Next, to know the order of recruitment of SET-Nup214 and CRM1 onto *HOX* clusters, we performed knockdown of SET-Nup214 in LOUCY cells. Reduction of SET-Nup214 caused down-regulation of *HOX* genes in agreement with a previous report (*Van Vlierberghe et al., 2008*), but did not significantly affect the amount of CRM1 that bound to *HOX* regions, suggesting that CRM1 functions as a recruiter of SET-Nup214 (*Figure 3—figure supplement 3*).

In addition, we performed this experiment using LMB. Unlike KPT-330, LMB irreversibly binds to CRM1 (*Sun et al., 2013*) and does not induce the degradation of CRM1 protein (*Turner et al., 2013*; *Mendonca et al., 2014*). Interestingly, we found that SET-Nup214 was also degraded by LMB treatment (*Figure 3—figure supplement 4A*), as by KPT-330 treatment (*Figure 3A*, *Figure 3—figure supplement 5A*). ChIP-qPCR revealed that LMB caused significant depletion of *HOX* cluster-bound CRM1 and SET-Nup214, and down-regulation of *HOX* genes (*Figure 3—figure supplement 4B–E*). These results demonstrate that CRM1/Ran-GTP/NES protein complex, which shows a higher affinity toward Nup214 or SET-Nup214 than CRM1 alone (*Askjaer et al., 1999*; *Kehlenbach et al., 1999*; *Port et al., 2016*), is important for the recruitment of SET-Nup214 to *HOX* clusters, and its stabilization. However, it is currently unknown which NES-containing protein(s) would be co-recruited with CRM1 and SET-Nup214 on the *HOX* genes.

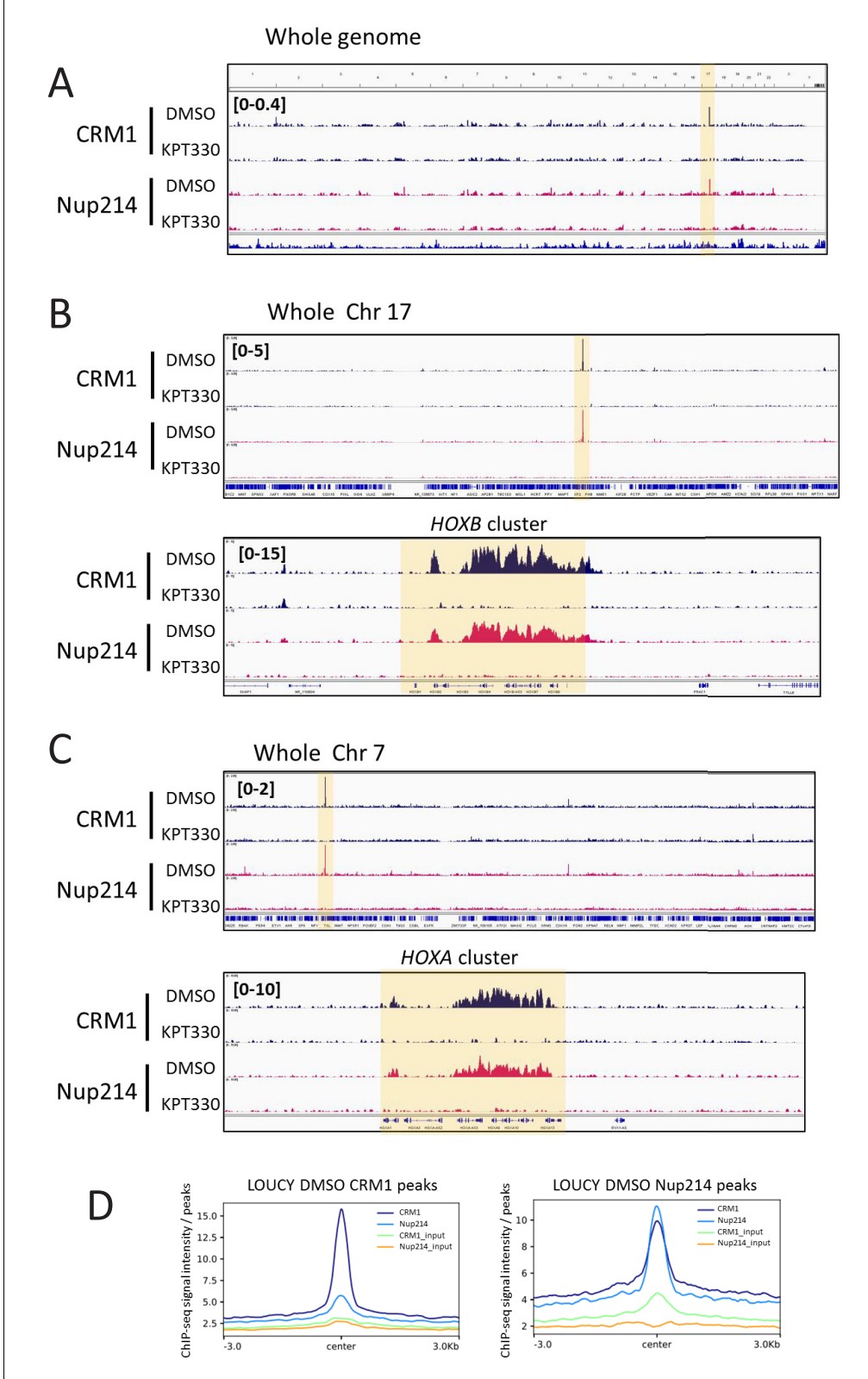

**Figure 2.** Accumulation of CRM1 and SET-Nup214 on *HOX* clusters is sensitive to CRM1 inhibitor KPT-330. (**A–C**) The binding profiles of CRM1 and Nup214 in LOUCY cell line treated either in the presence of DMSO (vehicle control) or KPT-330 (1000 nM) for 24 hr. (**A**: whole genome, **B**: whole chromosome 17 and *HOXB* cluster, **C**: whole chromosome seven and *HOXA* cluster). (**D**) Aggregation plots of CRM1 and SET-Nup214 binding sites in

*Figure 2 continued on next page*

*Figure 2 continued*

LOUCY. Nup214 binding signals in control (DMSO-treated) cells are mapped against CRM1 binding sites (left panel), and CRM1 binding signals are mapped against Nup214 binding sites (right panel).

The online version of this article includes the following figure supplement(s) for figure 2:

**Figure supplement 1.** Accumulation of CRM1 and Nup214 on specific genome regions is sensitive to CRM1 inhibitor KPT-330.

## SET-Nup214 and CRM1 recruit active Pol II onto the *HOX* clusters

SET, which was originally identified as a fusion gene with Nup214 (also known as CAN) (*von Lindern et al., 1992b*), possesses a homology with the yeast nucleosome assembly protein NAP-I and stimulates the replication of adenovirus core particles (*Matsumoto et al., 1993*). It has been further shown that SET is a multi-functional protein involved in histone chaperoning (*Kawase et al., 1996*; *Muto et al., 2007*), DNA repair (*Kalousi et al., 2015*), inhibition of protein phosphatase 2A (*Li et al., 1996*), and inhibition of histone acetyltransferase (*Miyamoto et al., 2003*; *Kutney et al., 2004*).

Therefore, we next tried to investigate the mechanism by which the accumulation of SET-Nup214 triggers robust *HOX* gene activation. Since it has been shown that SET is essential for transcription by RNA polymerase Pol II (Pol II) in vitro (*Gamble et al., 2005*; *Gamble and Fisher, 2007*), we speculated that the SET-Nup214 accumulation onto the *HOX* clusters may cause vigorous activation of Pol II. As expected, both phosphorylated Ser2 (S2P) and Ser5 (S5P) RNA Pol II significantly accumulated in the *HOXA* and *HOXB* clusters in both LOUCY and K562 cells, while almost no signals were observed in HL60 cells, which expressed quite low levels of *HOX* genes (*Figure 4A*). Moreover, we found that the levels of active RNA Pol II, S2P and S5P, on the *HOXA* and *HOXB* clusters were more sensitive to treatment with KPT-330 in the LOUCY cells than in K562 cells (*Figure 4B*). We further examined the profile of phosphorylated RNA Pol II (S2P or S5P) in LOUCY cells by ChIP-seq, and it was revealed that both S2P and S5P of the majority of *HOXA* and *HOXB* clusters were sensitive to KPT-330, while those in the surrounding regions were not affected (*Figure 4C–D*). Indeed, the genome-wide analysis of CRM1-bound genes revealed that both S2P and S5P of *HOXA* and *HOXB* were highly sensitive to KPT-330 (*Figure 4E*). Thus, these results indicate that the activation of the *HOX* cluster genes in LOUCY cells is highly dependent on the presence of CRM1 on the *HOX* clusters, suggesting that CRM1 together with SET-Nup214 creates a specific environment for robust *HOX* gene activation in LOUCY cells.

Notably, while our data show the binding of CRM1/SET-Nup214, in most of the cases, covering the whole region of *HOX* gene (*Figure 4—figure supplement 1*), the gene expression/active RNA Pol II shows differential sensitivity against KPT-330 among the *HOX* cluster regions. We speculate that, since various chromatin-bound proteins that are not evenly positioned are known to exist at *HOX* regions, these proteins could either positively or negatively affect SET-Nup214/CRM1-mediated *HOX* gene activation. As a result, some *HOX* genes, such as *HOXA11* and *HOXB9*, may be highly dependent on SET-Nup214/CRM1 for their activation [as observed in LOUCY cells (*Figure 3B–C*), but not in MEGAL cells (*Figure 3—figure supplement 2C–D*)], while others, such as *HOXA9* and *HOXB4*, may not.

The interaction between nucleoporin and nuclear transport receptor is important for active nuclear transport processes. The interactions between Nup98-CRM1 and Nup214-CRM1 have also been implicated during nuclear transport process (*Bogerd et al., 1998*; *Radu et al., 1995*; *Powers et al., 1997*; *Oka et al., 2010*; *Takeda et al., 2010*; *Hutten and Kehlenbach, 2006*; *Bernad et al., 2006*; *Port et al., 2015*; *Roloff et al., 2013*; *Saito et al., 2016*). However, our results suggest that ectopic interaction between these nucleoporins and CRM1 on the genomic DNA could trigger abnormal gene regulation.

## NPM1 mutant is also targeted to *HOX* cluster region in a CRM1-dependent manner

Our results so far indicate that CRM1 is important for the regulation of *HOX* genes. CRM1 is a nuclear export factor that binds to both nucleoporin and NES-containing substrates, suggesting that NES-containing proteins could be recruited to *HOX* clusters to regulate *HOX* genes. Certainly, a previous study suggested that CALM-AF10, an NES-containing fusion protein found in immature acute

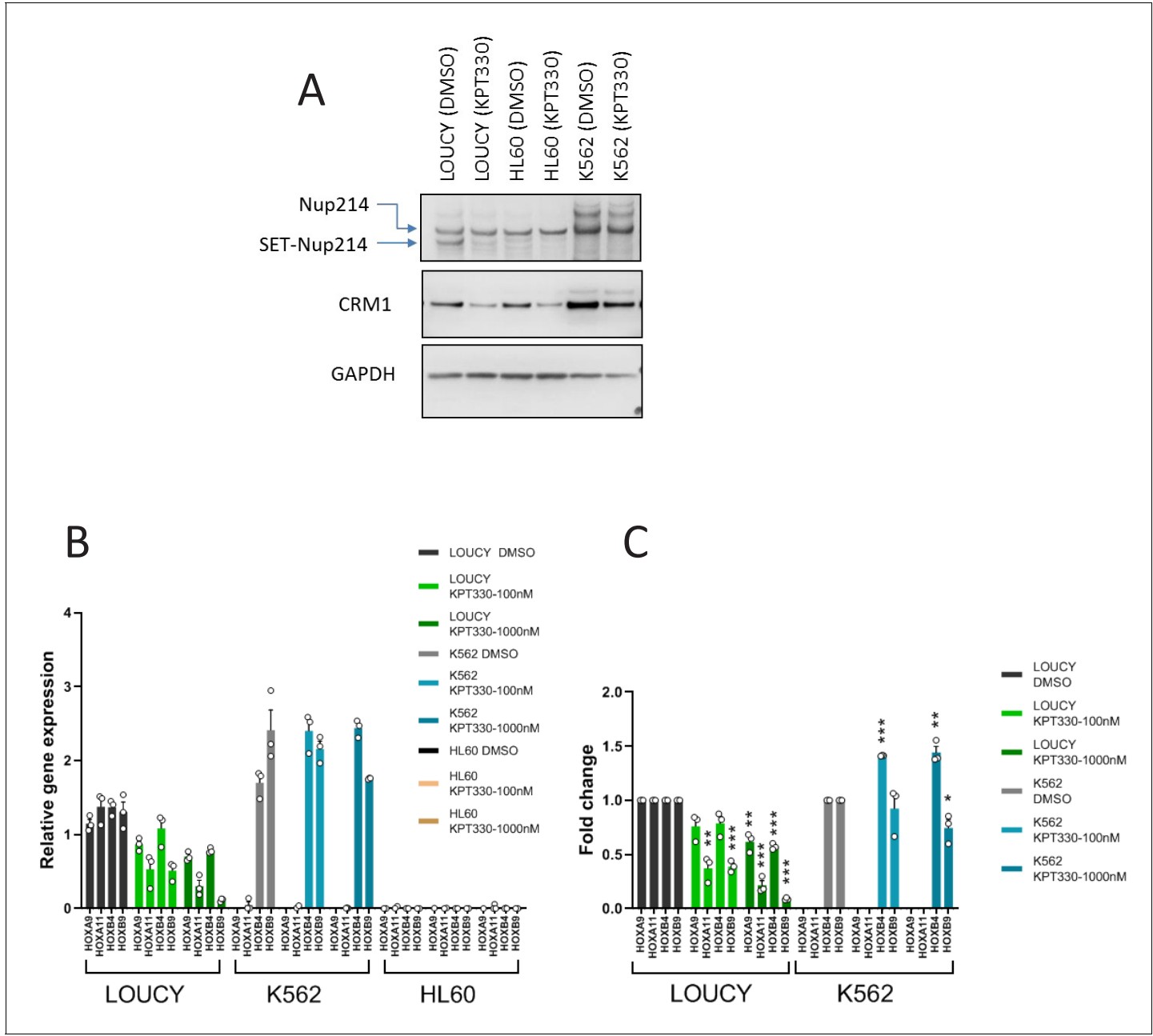

**Figure 3.** The expression of *HOX* genes in LOUCY cells are sensitive to CRM1 inhibitor KPT-330. (**A**) Protein expression of Nup214, SET-Nup214, CRM1, and GAPDH. LOUCY, HL60, and K562 cell lines were cultured either with DMSO (vehicle control) or KPT330 (1000 nM) for 24 hr; cell lysates were prepared by boiling the cells in a sample buffer and analyzed by immunoblotting using anti-Nup214, anti-CRM1, or anti-GAPDH antibodies. (**B**) qPCR analysis of *HOX* cluster genes (*HOXA9*, *HOXA11*, *HOXB4*, and *HOXB9*) in LOUCY, K562, and HL60 cell lines treated with DMSO (vehicle control) or KPT-330 (100 nM or 1000 nM) for 24 hr. *GAPDH* was used as a reference gene. Data are presented as mean values ± SEM of three independent experiments (n = 3). (**C**) The data in (**B**) were reanalyzed for the ratio as compared with the value for DMSO treated samples. Data are presented as mean values ± SEM. Low expressed genes (*HOXA9* and *HOXA11*) in K562 cells were omitted in (**C**). Asterisks indicate statistical significance determined by Student's *t*-test; *p<0.05; **p<0.01; ***p<0.001.

The online version of this article includes the following figure supplement(s) for figure 3:

**Figure supplement 1.** The nuclear dot structure in LOUCY cells are sensitive to CRM1 inhibitor KPT-330.

**Figure supplement 2.** The effect of KPT-330 on gene expression, and binding of CRM1 to *HOX* regions in MEGAL and HEL cells.

**Figure supplement 3.** Effect of SET-Nup214 knockdown.

**Figure supplement 4.** Effect of LMB on the accumulation of SET-Nup214/CRM1 onto *HOX* regions and gene activation.

**Figure supplement 5.** Protein expression and CRM1 binding profiles in HL60 cells.

myeloid and T-lymphoid malignancies, could be recruited to CRM1 (bound to transcription start site of *HOXA9* and *HOXA10*) in an NES-dependent manner (*Conway et al., 2015*). The same scenario could be observed in leukemia cells expressing the NPM1 mutant NPM1c, which contains a novel NES at its C-terminus and is the most frequent mutation in normal karyotype AML (*Falini et al., 2005*; *Verhaak et al., 2005*). It is important to note that NPM1c-expressing leukemia is known to be associated with robust *HOX* gene activation (*Alcalay et al., 2005*; *Mulligan et al., 2007*; *Spencer et al., 2015*). Furthermore, it has been shown that the abrogation of NES of NPM1c in OCI-AML3, an AML cell line that expresses NPM1c, causes a robust downregulation of *HOX* gene expression (*Brunetti et al., 2018*). Thus, it is most likely that NPM1c can bind to chromatin-bound CRM1 via its NES (NES-CRM1 interaction) like Nup98- or Nup214-fusions, although Nup98 and Nup214 bind to CRM1 via their FG repeats (nucleoporin-CRM1 interaction).

We next examined whether NPM1c binds to the *HOX* cluster regions using OCI-AML3 cells. We performed ChIP-seq analysis using an antibody specific for NPM1c. Specificity of anti-NPM1c antibody was confirmed using stable cell lines expressing FLAG-tagged NPM1 or NPM1c (*Figure 5A*). We found that NPM1c robustly accumulated in *HOXA* and *HOXB* cluster regions (*Figure 5B*). We also found that, in OCI-AML3, CRM1 bound to both the *HOXA* and *HOXB* clusters, in a similar fashion as that to NPM1c (*Figure 5B*). Aggregation plots revealed moderate enrichment of NPM1c around CRM1 binding sites (*Figure 5C*). This relatively low enrichment as compared with that of SET-Nup214 suggested that NPM1c could also bind to various genomic regions, independent of CRM1.

In agreement with a previous study (*Brunetti et al., 2018*), the expression of *HOX* genes was sensitive to KPT-330 in OCI-AML3 cells (*Figure 5—figure supplement 1B–C*). Furthermore, treatment with KPT-330, which reduced the protein level of CRM1 but not that of NPM1c (*Figure 5—figure supplement 1A*), caused significant reduction of both CRM1 and NPM1c bound to chromatin on *HOX* cluster genes (*Figure 5D and E*). In addition, both S2P and S5P forms of active Pol II showed significant sensitivity toward KPT330 treatment, as observed in SET-Nup214 expressing LOUCY cells (*Figure 5—figure supplement 2*). The robust signal of active Pol II was observed at *HOXA9* region, which correlated well with CRM1 and NPM1c binding (*Figure 5D*). Furthermore, previous RNA-seq data and ChIP-seq data of active histone marks in OCI-AML3 cells (*Brunetti et al., 2018*) correlated well with the binding pattern of CRM1/NPM1c (*Figure 5—figure supplement 3*). Especially, the binding signals of NPM1c/CRM1 and active histone marks were enriched in posterior (AbdB-related) *HOX* genes in the *HOXA* cluster (*HOXA9* to *HOXA13*). Collectively, NPM1c is most likely involved in the activation of Pol II together with CRM1 in OCI-AML3.

These results demonstrate that NPM1c and CRM1 co-accumulate onto the *HOX* cluster regions to activate genes in OCI-AML3 cells.

To further examine whether NPM1c could be recruited to chromatin-bound CRM1, we next utilized mouse embryonic-stem (ES) cell lines. Our previous study demonstrated that CRM1 robustly accumulates in all four *Hox* cluster regions (*HoxA*, *HoxB*, *HoxC*, and *HoxD*) in mouse ES cells (*Oka et al., 2016*). In contrast, in this study, our data showed that CRM1 binds to only two *HOX* cluster regions (*HOXA* and *HOXB*) in OCI-AML3 cells (*Figure 5B*). Therefore, NPM1c could be recruited to the four *Hox* cluster regions in ES cells if it could be recruited by chromatin-bound CRM1.

In a previous study, we used polyclonal anti-CRM1 antibody for ChIP-seq to determine genome-wide binding sites of CRM1 (*Oka et al., 2016*). Therefore, to further confirm the binding of CRM1 to the *Hox* cluster regions, we established an ES cell line expressing FLAG-tagged endogenous CRM1. By using Crisper/Cas9–mediated gene editing, we successfully introduced a monoallelic insertion to express C-terminal FLAG-tagged CRM1 (Clone#2–34; *Figure 6A–C*). ChIP-sequencing analysis with anti-FLAG antibody revealed high peaks of signals on the four *Hox* cluster regions, similar to those of the CRM1 binding profile using polyclonal anti-CRM1 antibody (*Figure 6D*, *Figure 6—figure supplement 1*), confirming that CRM1 actually is associated with these regions of chromatin in mouse ES cells.

During characterization of the gene-edited cell lines, we noticed that there was a clone (#2–2) containing a four amino acids deletion (ΔHEIP) at the C-terminus of CRM1, a region which has been shown to be important for autoinhibition (*Dong et al., 2009*; *Saito and Matsuura, 2013*; *Fox et al., 2011*) (*Figure 6A–C*). Immunoblotting with anti-CRM1 (CST; #46249), which was raised against the C-terminus of human CRM1, only detected CRM1-FLAG protein in #2–34 cells, while anti-CRM1

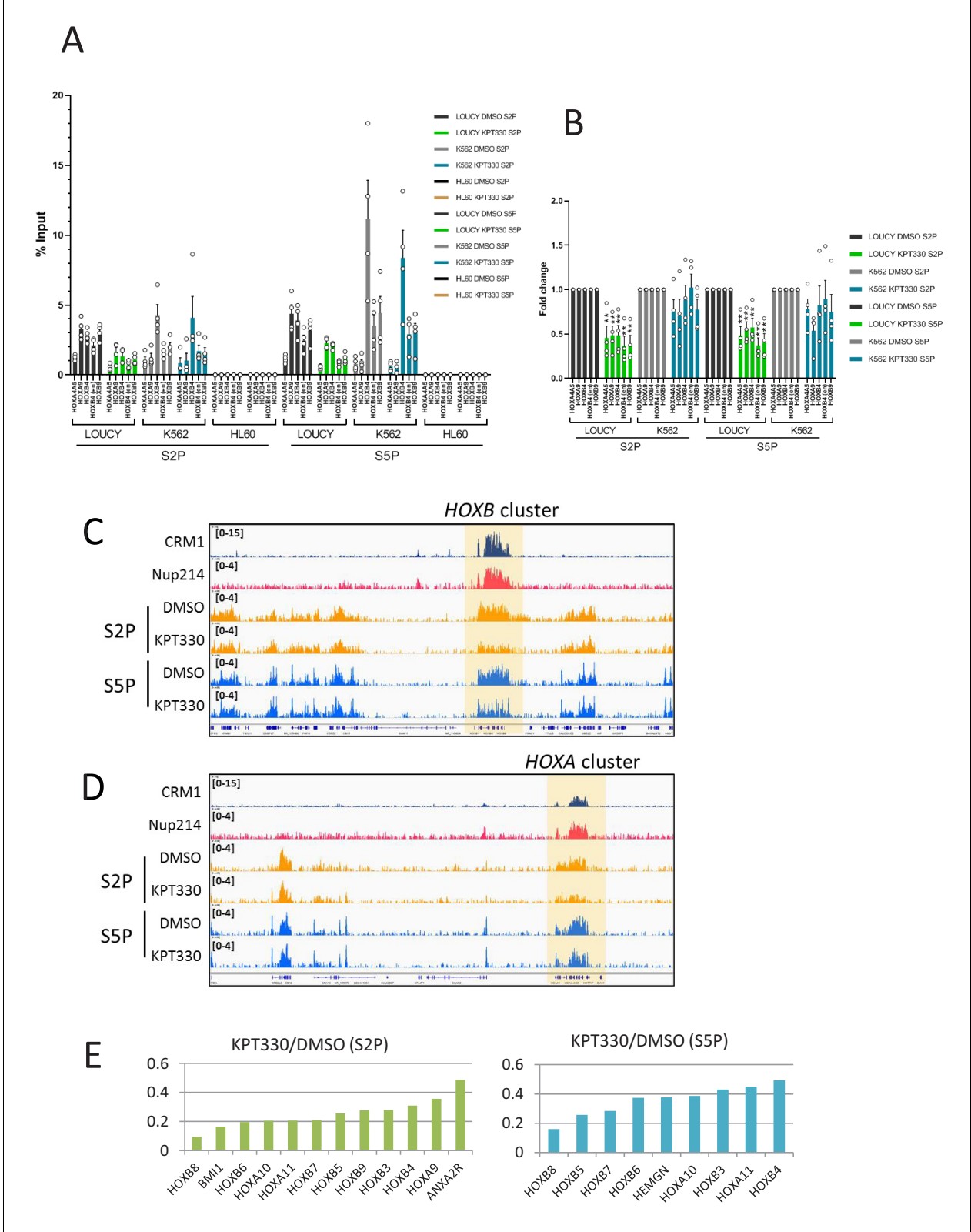

**Figure 4.** *HOX* cluster region shows CRM1-dependent activation of RNA Pol II in LOUCY cells. (**A**) ChIP-qPCR analysis of active RNA Polymerase II (Pol II) (Ser2- or Ser5-phosphorylated) at *HOX* gene loci in LOUCY, K562, and HL60 cell lines cultured either in the presence of DMSO (vehicle control) or KPT-330 (1000 nM) for 24 hr. The primer set used was as follows: HOXA4A5 (intergenic region between *HOXA4* and *HOXA5*); *HOXA9* (promoter); *HOXB4* (promoter); *HOXB4* (enhancer); *HOXB9* (promoter). Data are presented as mean values ± SEM of four independent experiments (n = 4). (**B**) The

Figure 4 continued

data in (A) were reanalyzed to obtain a ratio by comparing with the value for DMSO treated samples. Data are presented as mean values ± SEM. Asterisks indicate statistical significance determined by Student's t-test; **p<0.01. (C–D) Binding profiles of Ser2- or Ser5-phosphorylated Pol II at HOXB (C) or HOXA (D) clusters cultured either in the presence of DMSO (vehicle control) or KPT-330 (1000 nM) for 24 hr. (E) The most affected genes (among CRM1-bound genes) by the treatment with KPT-330. S2P (left panel) and S5P (right panel).

The online version of this article includes the following figure supplement(s) for figure 4:

**Figure supplement 1.** ChIP-seq profiles of CRM1, SET-Nup214, S2P (DMSO, KPT-330), S5P (DMSO, KPT-330) around HOXA9, HOXA11, HOXB4, and HOXB9 genes.

(Santa Cruz; H-300; raised against a. a. 772–1071 region of human CRM1) or anti-CRM1 (BD; #611832; raised against a. a. 2–122 region of human CRM1) detected the CRM1-FLAG protein in both #2–2 and #2–34, confirming the deletion of the C-terminus of CRM1 in clone #2–2 (*Figure 6B*).

We found that this ΔHEIP mutant (clone #2–2) showed a somewhat enhanced binding to nuclear envelope compared to that of a clone without deletion (clone #2–34) (*Figure 6C*). ChIP-seq analysis using FLAG antibody demonstrated that ΔHEIP mutant also preferentially bound to four *Hox* cluster regions (*Figure 6D*, *Figure 6—figure supplement 1*). Together, these results confirm that CRM1 indeed bound to the *Hox* cluster regions in ES cells.

Next, ES cell lines that ectopically express FLAG-tagged NPM1c or, as a control, wild-type NPM1, were established (*Figure 7A*). As expected, we observed a robust accumulation of NPM1c, but not wild-type NPM1, in all four *Hox* cluster regions (*Figure 7B*). These results clearly indicate that NPM1c is targeted to the *HOX* clusters in a NES-CRM1 interaction–dependent manner in the nucleus. However, gene expression analysis revealed that NPM1c could not activate *Hox* gene in ES cells (data not shown). We speculate that *Hox* cluster region in ES cells may possess a different chromatin structure or epigenetic status as compared with that of OCI-AML3 cells and may be difficult to be activated solely by the ectopic expression of NPM1c. Further study will be needed to determine the exact mechanism of gene activation by NPM1c.

Collectively these results demonstrate that CRM1 creates a platform on the *HOX* cluster regions to recruit the leukemogenic proteins, either via nucleoporin-CRM1 (Nup98- or Nup214-fusions to CRM1) interaction or NES-CRM1 (NPM1c to CRM1) interaction, to induce aberrant expression of *HOX* genes (*Figure 8*).

Recently, it has been elegantly shown that *HOX* genes are not indirectly activated by NPM1c but that NPM1c more directly causes the upregulation of *HOX* genes for the maintenance of a stem cell–like state (*Brunetti et al., 2018*). That is, the inactivation of NES or degradation of NPM1c causes a rapid downregulation of specific subsets of genes important for leukemogenesis, including *HOXA9*, *HOXA11*, *HOXB3*, *HOXB8*, *MEIS1*, and *CDKN2C* (*Brunetti et al., 2018*), although the mechanism by which NPM1c activates *HOX* genes remains unknown. Consistently, in this study, we found that NPM1c is targeted to the *HOX* cluster regions, showing the direct contribution of NPM1c on *HOX* gene activation.

Our observation that NPM1c binds to the *HOX* cluster regions seems to be clinically significant. Ongoing therapeutic strategies that target the function of CRM1 are focused on the inhibition of the nuclear export process of disease-related NES-containing cargoes, including tumor suppressor proteins. However, it should be taken into consideration that the export of various NES-containing cargoes, essential for normal cells to survive, may be also affected by such strategy, since it has been demonstrated that CRM1 binds to and exports a diverse set of proteins involved in various essential cellular processes (*Kirli et al., 2015*; *Thakar et al., 2013*). Thus, our results imply the possibility that chromatin-bound CRM1 could be another therapeutic target in leukemia with high expression of *HOX* genes in a CRM1-dependent manner, although it still remains to be known as to how and why CRM1 accumulates in the *HOX* cluster regions, and the chromatin-bound CRM1 may play other physiological roles in normal cells.

In conclusion, we have now demonstrated that three representative leukemogenic proteins, Nup98-fusion, Nup214-fusion, and NPM1 mutant, are recruited to *HOX* cluster regions through their interaction with CRM1, pre-bound to the *HOX* cluster regions, to specifically activate *HOX* genes.

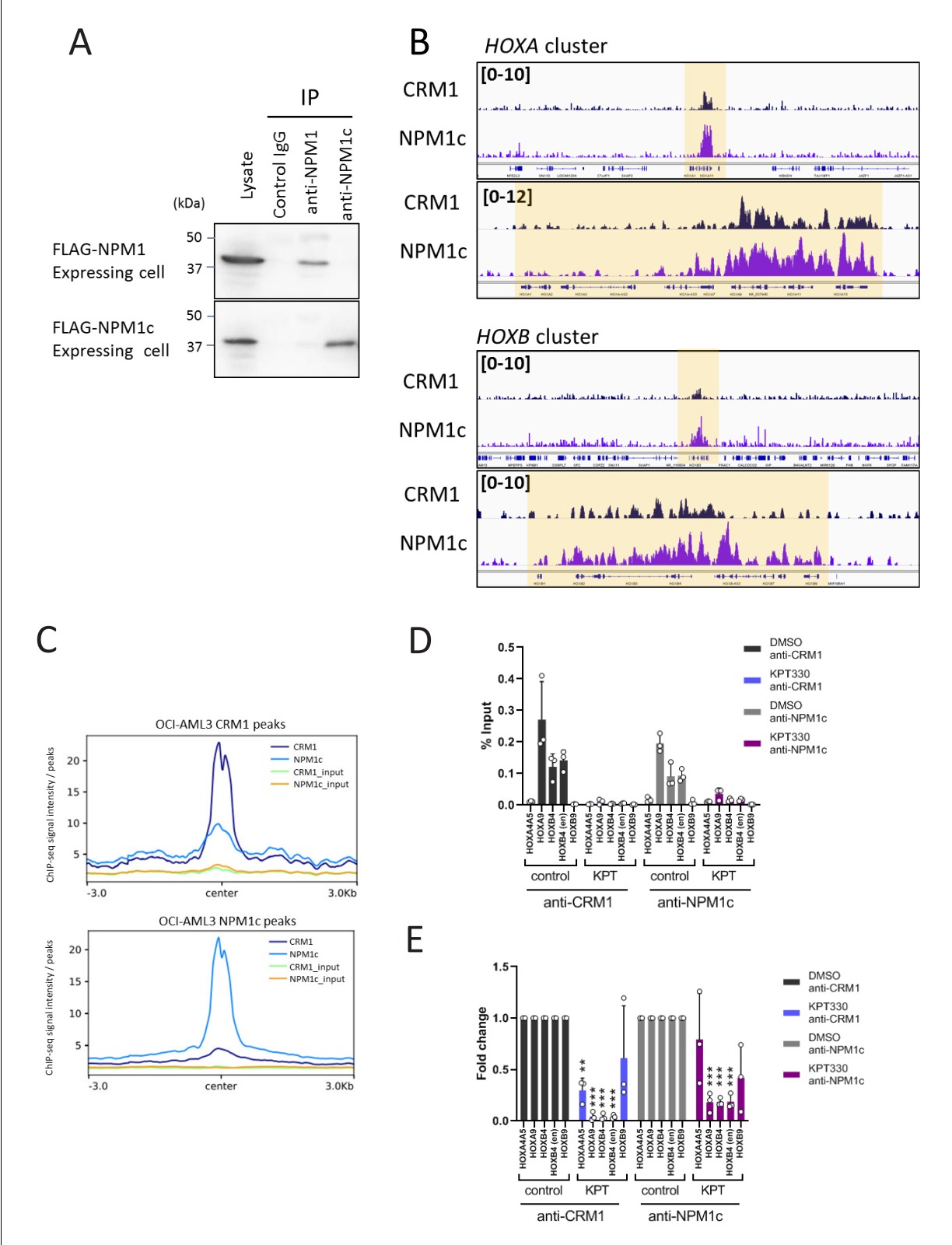

**Figure 5.** NPM1 mutant NPM1c accumulated onto *HOX* clusters together with CRM1. (**A**) The specificity of anti-NPM1c or anti-NPM1 antibodies was demonstrated by immunoprecipitation. Cell lysates from mouse ES cells stably expressing either FLAG-NPM1 or FLAG-NPM1c were used for immunoprecipitation using indicated antibodies and analyzed by immunoblotting using anti-FLAG antibody. (**B**) Binding profiles of CRM1 and NPM1c in OCI-AML3 cell line (*HOXA* and *HOXB* clusters). ChIP-seq was performed using anti-CRM1 or anti-NPM1c antibodies. (**C**) Aggregation plots of CRM1
*Figure 5 continued on next page*

*Figure 5 continued*

and NPM1c binding sites in OCI-AML3 cells. NPM1c binding signals in control (DMSO-treated) cells are mapped against CRM1 binding sites (top), and CRM1 binding signals of DMSO-treated cells are mapped against NPM1c binding signals (bottom). (D) ChIP-qPCR analysis of CRM1 and NPM1c at *HOX* gene loci in OCI-AML3 cells cultured either in the presence of DMSO (vehicle control) or KPT-330 (100 nM) for 24 hr. The primer set used was as follows: *HOXA4A5* (intergenic region between *HOXA4* and *HOXA5*); *HOXA9* (promoter); *HOXB4* (promoter); *HOXB4* (enhancer); *HOXB9* (promoter). Data are presented as mean values ± SEM of three independent experiments (n = 3). (E) The data in (D) were reanalyzed to obtain a ratio by comparing with the value for DMSO-treated samples. Data are presented as mean values ± SEM of three independent experiments (n = 3). Asterisks indicate statistical significance determined by Student's *t*-test; **p<0.01; ***p<0.001.

The online version of this article includes the following figure supplement(s) for figure 5:

**Figure supplement 1.** The effect of KPT-330 on the protein and *HOX* gene expression levels in OCI-AML3 cells.
**Figure supplement 2.** The effect of KPT-330 on active Pol II at *HOX* loci in OCI-AML3 cell line.
**Figure supplement 3.** ChIP-seq data in this study (CRM1 and NPM1c in OCI-AML3 cells) were analyzed for comparison with RNA-seq and ChIP-seq data in OCI-AML3 (*Brunetti et al., 2018*).

# Materials and methods

## Cell lines and cell culture

Leukemia cell lines, LOUCY, MEGAL, and OCI-AML3 were purchased from DSMZ (Braunschweig, Germany). K562, HL60, and HEL cells were obtained from JCRB cell bank (Ibaraki, Osaka). These cell lines were authenticated using STR typing by DSMZ or JCRB cell bank, and also the mycoplasma status was checked by them. LOUCY, MEGAL, and HL60 cells were cultured in RPMI1640 medium (Sigma) supplemented with 20% FBS (Sigma). K562 and HEL cells were cultured in RPMI1640 medium supplemented with 10% FBS. OCI-AML3 was cultured in alpha-MEM medium (WAKO) supplemented with 20% FBS. EB3 ES cells (*Niwa et al., 2002*; *Ogawa et al., 2004*), E14tg2a ES cells (*Hooper et al., 1987*) and their derivatives, and HeLa cells were cultured as described previously (*Oka et al., 2016*).

## IP-immunoblotting

To ensure the specificity of anti-NPM1c antibody, we performed IP-immunoblotting. Briefly, mouse ES cells stably expressing FLAG-tagged NPM1 or NPM1c were resuspended in RIPA buffer [50 mM Tris-Cl pH 8.0, 150 mM NaCl, 1% NP40, 0.5% Na-deoxycholate, 0.1% SDS, 1 mM EDTA, protease inhibitor cocktail (cOmplete ULTRA; Roche)], and sonicated. After centrifuging at 20,400 × *g* for 10 min at 4°C, 1 µg of antibody was added to the cell lysate and incubated overnight. Later, 20 µl of Protein G Sepharose beads (GE Healthcare) were added and the samples incubated for 1 hr at 4°C. After washing with RIPA buffer five times, the bound proteins were eluted in sample buffer for 5 min at 95°C. Finally, immunoblotting with anti-NPM1 or anti-NPM1c antibody was performed.

## Plasmids, transfection, and generation of stable clones

Generation of stable ES cell lines expressing a transgene using *Tol2* transposon vector (*Kawakami and Noda, 2004*; *Urasaki et al., 2006*) was performed as described previously (*Oka et al., 2013*). Picked colonies were expanded to confirm the expression of the transgene by immunoblotting and immunofluorescence staining. cDNA encoding for human NPM1 was PCR amplified using a cDNA library prepared from HeLa cells. An NPM1 mutant, NPM1c, was created via PCR using long primer coding mutant sequences. Either the wild-type NPM1 or NPM1c was cloned into BamHI-NotI site of pT2A-CMH-FLAGx3 (*Oka et al., 2013*).

ES3 cells were co-transfected with the plasmids pCAGGS-m2TP (an expression vector for *Tol2* transposase) and pT2A-CMH-FLAGx3-NPM1 or pT2A-CMH-FLAGx3-NPM1c using lipofectamine 2000 (Life Technologies). After 2 days, cells were re-plated onto the ES-LIF medium containing hygromycin B (200 µg/ml). Colonies were picked and examined for the expression of FLAG-NPM1 or FLAG-NPM1c by immunoblotting and immunofluorescence staining.

## Immunofluorescence staining and confocal microscopy

For leukemia cell lines, cells were transferred to Eppendorf tubes and fixed in a medium containing 3.7% formaldehyde for 15 min at room temperature. Cells were collected by centrifuge at 800 × *g* for 5 min at room temperature. After a wash with PBS, cells were permeabilized with 0.5% Triton

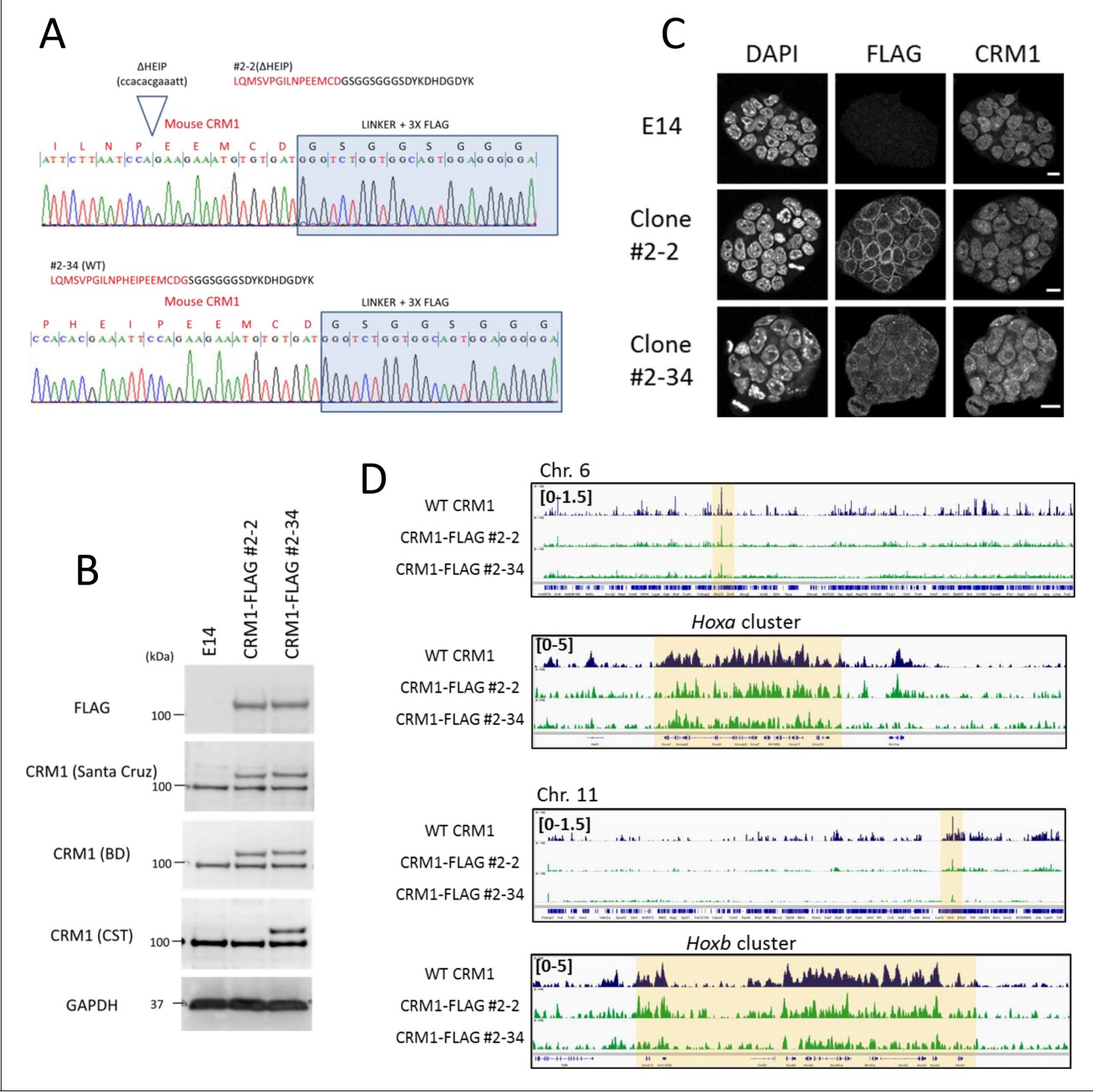

**Figure 6.** Characterization of the cell lines that express C-terminal FLAG-tagged endogenous CRM1 generated by genome editing. (**A**) Genome sequence analysis of FLAG-tag knocked-in cells (ΔHEIP [clone#2–2] or WT [clone#2–34; no deletion]). (**B**) Immunoblotting analysis of cell lines using anti-FLAG, anti-CRM1, or anti-GAPDH antibodies. (**C**) Immunofluorescent analysis of cell lines using anti-FLAG and anti-CRM1 antibodies. Scale bars: 10μm. (**D**) ChIP-seq analysis of cell lines using anti-CRM1 (EB3; E14 derived ES; dark blue) or anti-FLAG antibody (FLAG-knocked-in ES cells; #2-2[ΔHEIP] or #2–34 [no deletion]). Whole chromosome six and *HoxA* cluster regions, or whole chromosome 11 and *HoxB* cluster regions are shown. The online version of this article includes the following figure supplement(s) for figure 6:

**Figure supplement 1.** ChIP-seq analysis of indicated cell lines.

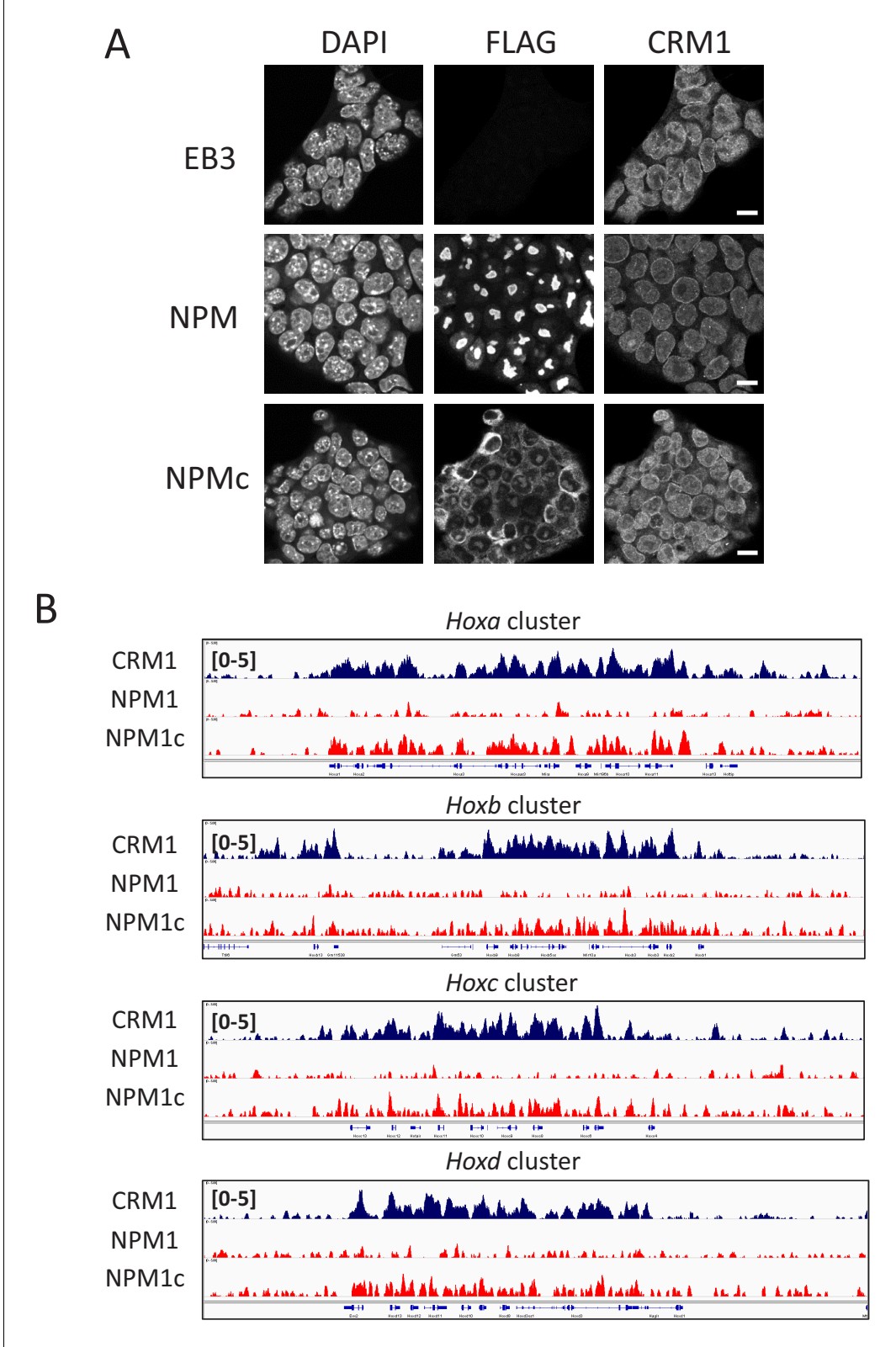

**Figure 7.** NPM1c binds to four *Hox* cluster regions in mouse ES cells. (**A**) Immunofluorescent analysis of stable cell lines (EB3 parental ES cells, FLAG-NPM1 expressing ES cells, or FLAG-NPM1c expressing ES cells) using anti-FLAG or anti-CRM1 antibody. Scale bars: 10μm. (**B**) ChIP-seq analysis of cell lines indicated using anti-FLAG antibody. All four *Hox* cluster regions (*HoxA*, *HoxB*, *HoxC*, and *HoxD*) are shown. (Top) dark blue, anti-CRM1 ChIP

*Figure 7 continued on next page*

Figure 7 continued

signal; (middle) red, anti-FLAG ChIP signal for FLAG-NPM1 expressing ES cells; (bottom) red, anti-FLAG ChIP signal for FLAG-NPM1c expressing ES cells.

X-100 in PBS for 5 min at room temperature. Cell were washed in PBS and incubated with blocking buffer (3% skim milk in PBS) for 30 min with rotation. Afterwards, cells were incubated with primary antibodies overnight at 4°C with rotation. After washing three times with PBS, cells were incubated with secondary antibodies for 30 min. Cells were washed with PBS twice and stained with DAPI in PBS for 10 min at room temperature. Cells were collected as a pellet, resuspended in ProLong Gold (Thermo Fischer Scientific), and mounted. Images were acquired using an SP8 confocal microscope (Leica).

ES cells were grown on coverslips and fixed in a medium containing 3.7% formaldehyde for 15 min at room temperature. After treatment with 0.5% Triton X-100 in PBS for 5 min, the cells were incubated in a blocking buffer (3% skim milk in PBS) for 30 min. Next, cells were incubated with primary antibodies overnight at 4°C. After washing with PBS four times, cells were incubated with secondary antibodies for 30 min. Next, the cells were washed with PBS four times, stained with DAPI for 10 min at room temperature, and the coverslips were mounted with ProLong Gold (Thermo Fischer Scientific). Images were acquired using the SP8 confocal microscope (Leica).

## Immunoblotting

Cells were collected by centrifugation and RIPA buffer containing a protease inhibitor cocktail (cOmplete ULTRA, Roche) was added. Cells were then vortexed, and incubated on ice for 30 min. After centrifugation at 20,400 $\times$ g for 10 min, supernatant was collected. Protein concentration of each sample was measured by BCA protein assay kit (Pierce). In *Figure 3A*, sample buffer was directly added to the cell pellet and boiled for 10 min. After separation of proteins (15 µg in *Figure 3A—figure supplement 5*, 10 µg in other Figures) on SDS-PAGE gels, proteins were transferred to a polyvinylidene fluoride membrane (Immobilon-P; Millipore) and probed using mouse anti-FLAG M2 (Sigma), anti-CRM1 (CST, SnataCruz, or BD), anti-Nup214 (Bethyl), or anti-GAPDH (MBL) antibody. Later, the membrane was incubated with secondary antibody. Signals were detected by Chemi-Lumi One Super (Nacalai).

## qPCR

Total RNA was extracted from cells using ReliaPrep RNA Miniprep Systems (Promega) and used for cDNA synthesis with the PrimeScript RT reagent Kit (Takara Bio). All procedures were conducted according to the manufacturer's recommendations. qPCR analysis was performed on a 384-well plate with QuantStudio 6 Flex Real-Time PCR System (Life Technologies) using GeneAce SYBR qPCR Mix (Nippon gene). Relative gene expression levels were normalized using *GAPDH* mRNA levels as control. The primer sequences are listed in *Supplementary file 1*.

## ChIP analysis

Leukemia cells were fixed in a medium containing 0.5% formaldehyde at room temperature for 5 min. Cells were collected by centrifuge at 800 $\times$ g for 5 min at room temperature. After washing with ice-cold PBS twice, cells were resuspended in the ChIP buffer (10 mM Tris-HCl pH 8.0, 200 mM KC), 1 mM CaCl$_2$, 0.5% NP40) containing protease inhibitors (2 µg/ml aprotinin, 2 µg/ml leupeptin, and 1 µg/ml pepstatin A) and briefly sonicated (Branson 250D Sonifier, Branson Ultrasonics). After centrifugation, the supernatants containing chromatin were digested with 1 U/ml micrococcal nuclease (Worthington Biochemical) for 40 min at 37°C, and the reaction was stopped with ethylene diamine tetraacetic acid (EDTA; final concentration of 10 mM). Enzyme-treated supernatants were incubated with anti-mouse or anti-rabbit IgG magnetic beads (Dynabeads, Life Technologies) preincubated with anti-FLAG M2 (Sigma), anti-CRM1 (CST or Bethyl lab), anti-Nup214 (Bethyl lab), anti-NPM1c (Thermo Scientific), or anti-Pol II (S2P), anti-Pol II (S5P) (*Odawara et al., 2011*) antibodies (2 µL) for 6 hr. [For Pol II ChIP, we first incubated the beads with rabbit anti-Rat IgG antibody (Jackson Immuno Research, West Grove, PA) before its incubation with phospho-specific Pol II antibodies.] The beads were washed twice thoroughly with each of the following buffers: ChIP buffer, ChIP wash

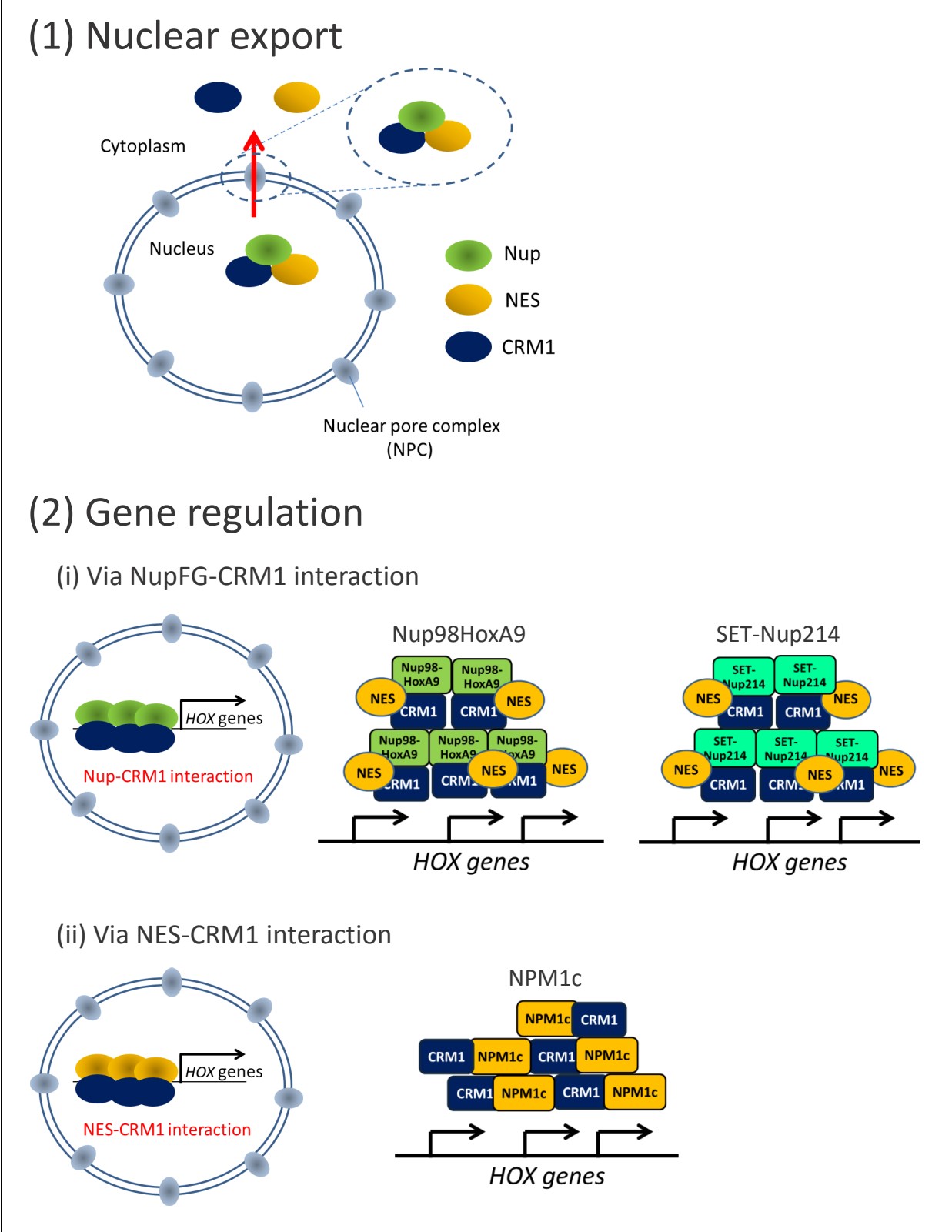

**Figure 8.** CRM1 functions as a platform to recruit leukemogenic proteins to *HOX* cluster regions. CRM1 possesses two distinct functions: (1) Nuclear export of NES-cargo proteins, and (2) gene regulation. Leukemogenic proteins co-localize with CRM1 at *HOX* cluster regions either via (i) CRM1-nucleoporin (FG) interaction (Nup98HoxA9 and SET-Nup214) or (ii) CRM1-NES interaction (NPM1c). Note that both the binding of these leukemogenic

*Figure 8 continued on next page*

*Figure 8 continued*

proteins to *HOX* cluster(s) and *HOX* gene expression are dependent on CRM1. In the case of Nup98HoxA9 and SET-Nup214, the NES-containing protein(s) is/are most likely involved in the complex.

buffer (10 mM Tris-HCl pH 8.0, 500 mM KCl, 1 mM CaCl$_2$, 0.5% NP40), and TE buffer (10 mM Tris-HCl pH 8.0, 1 mM EDTA), and eluted in an elution buffer containing 50 mM Tris-HCl pH 8.0, 10 mM EDTA, and 1% sodium dodecyl sulfate overnight at 65˚C. DNA was recovered using a DNA gel extraction kit (Promega) and was used for ChIP-qPCR analysis or preparation of library for ChIP-seq analysis.

### ChIP-seq and data analysis

The ChIP library was prepared using ThruPLEX DNA-seq kit (Takara Bio) according to the manufacturers' instructions and sequenced on the Illumina HiSeq1500 system. The sequence reads were uniquely aligned accordingly to the reference mouse genome (GRCm38) and human genome (GRCh38) using the HISAT software (version 2.1.0; *Kim et al., 2015*) with duplicates removed. For the ChIP-seq signal visualization for each sample, read counts in 100 bp bins smoothed over 100 bp windows were normalized by reads per kilobase per million (RPKM) using the *bamCoverage* program from the deepTools suite (version 3.1.3; *Ramírez et al., 2014*); *bamCompare* was used to subtract the corresponding input signal from the ChIP signal. Peaks were called with the software MACS wherein default parameters were employed for those samples involving S2P/S5P and cutoff was set at p<0.001 for the rest of the samples.

### CRISPR/Cas9 mediated C-terminus FLAG tagging of endogenous CRM1

The generation of cells containing C-terminus FLAG tagged endogenous CRM1 was performed as described (*Schmid-Burgk et al., 2016*). Briefly, E14 ES cells were transfected with pX330 containing an sgRNA targeting the C-terminus region of mouse CRM1 (target: 5'-ATTTCTTCTGGAATTTCGTG TGG-3'), together with pCAS9-mCherry (frame +0) and pCRISPAINT-3xFLAG-PuroR-mXPO1 (a scarless donor plasmid), using lipofectamine 2000. Two days after transfection, the cells were passaged into media containing puromycin (300 ng/ml). Among the growing colonies, a total of 36 colonies were picked, expanded, and examined for the expression of the transgene by immunoblotting and immunofluorescence analysis. Finally, correct in-frame insertion of FLAG (x3) was confirmed by sequence analysis.

### Knockdown of SET-Nup214 gene

LOUCY cells (1 × 10$^7$ cells) were transfected with 4 µM of SET siRNA (5'-ATGGAAATC TGGAAAGGAT-3') (*Anazawa et al., 2005*) or negative control siRNA [Universal Negative Control siRNA (Nippon gene)] by nucleofection (Lonza) using reagent V and the X-001 program. Cells were plated on 100 mm dish using RPMI 1640 medium supplemented with 20% FBS for the indicated period and used for immunoblotting and ChIP-qPCR.

### Accession numbers

The ChIP-Seq data are accessible through GEO Series accession number GSE127983.

### Statistical analysis

Statistical analyses were performed using an unpaired Student's two-sided *t*-test. Correction for multiple-comparisons was performed using the Holm-Sidak method. GraphPad Prism version eight was used for data analysis and representation.

## Acknowledgements

We thank Dr. Koichi Kawakami for *Tol2* transposon-based vectors, Dr. Hitoshi Niwa for ES cell lines, and Drs. Haruhiko Koseki, Shinsuke Ito, and Ichiro Hiratani for valuable discussions. This work was supported in part by JSPS KAKENHI grant numbers 16K14676 and 17H03679 (to MO), 25116008 and 16H04789 (to YY and MO), 19H05244, 18H05527, 18H04802, and 17H03608 (to YO), JST CREST

(JPMJCR16G1) (to YO), by a research grant from Japan Leukemia Research Fund (to MO) and Hoansha Foundation (to MO).

## Additional information

### Funding

| Funder | Grant reference number | Author |
| --- | --- | --- |
| Japan Society for the Promotion of Science | 16K14676 | Masahiro Oka |
| Japan Society for the Promotion of Science | 17H03679 | Masahiro Oka |
| Japan Society for the Promotion of Science | 25116008 | Masahiro Oka Yoshihiro Yoneda |
| Japan Society for the Promotion of Science | 16H04789 | Masahiro Oka Yoshihiro Yoneda |
| Hoansha Foundation | | Masahiro Oka |
| Japan Leukemia Research Fund | | Masahiro Oka |
| Japan Society for the Promotion of Science | 19H05244 | Yasuyuki Ohkawa |
| Japan Society for the Promotion of Science | 18H05527 | Yasuyuki Ohkawa |
| Japan Society for the Promotion of Science | 18H04802 | Yasuyuki Ohkawa |
| Japan Society for the Promotion of Science | 17H03608 | Yasuyuki Ohkawa |
| Japan Science and Technology Agency | MJCR16G1 | Yasuyuki Ohkawa |

The funders had no role in study design, data collection and interpretation, or the decision to submit the work for publication.

### Author contributions

Masahiro Oka, Conceptualization, Resources, Formal analysis, Supervision, Funding acquisition, Validation, Investigation, Writing—original draft, Project administration, Writing—review and editing; Sonoko Mura, Mayumi Otani, Yoichi Miyamoto, Kazumitsu Maehara, Akihito Harada, Validation, Investigation; Jumpei Nogami, Formal analysis, Validation, Investigation; Taro Tachibana, Resources; Yoshihiro Yoneda, Supervision, Funding acquisition, Writing—review and editing; Yasuyuki Ohkawa, Validation, Investigation, Project administration

### Author ORCIDs

Masahiro Oka (iD) https://orcid.org/0000-0003-0123-3060
Yasuyuki Ohkawa (iD) https://orcid.org/0000-0001-6440-9954

### Decision letter and Author response

Decision letter https://doi.org/10.7554/eLife.46667.SA1
Author response https://doi.org/10.7554/eLife.46667.SA2

## Additional files

### Supplementary files

• Supplementary file 1. List of primers used in this study.

• Supplementary file 2. Key resources table.

• Transparent reporting form

## Data availability

ChIP-seq data have been deposited in GEO under accession codes GSE127983.

The following dataset was generated:

| Author(s) | Year | Dataset title | Dataset URL | Database and Identifier |
|---|---|---|---|---|
| Oka M, Yoneda Y, Nogami J, Maehara K, Harada A, Oh-kawa Y | 2019 | Genome-wide mapping of SET-Nup214, mutant NPM1 (NPM1c), and Crm1-binding sites in human leukemia or mouse ES cell lines | https://www.ncbi.nlm.nih.gov/geo/query/acc.cgi?acc=GSE127983 | NCBI Gene Expression Omnibus, GSE127983 |

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
