## [Decision Letter]

**Decision letter after peer review:**

Thank you for submitting your article "Chromatin-bound CRM1 promotes recruitment of SET-Nup214 and NPM1c onto *HOX* clusters causing aberrant *HOX* expression" for consideration by *eLife*. Your article has been reviewed by two peer reviewers, and the evaluation has been overseen by a Reviewing Editor and James Manley as the Senior Editor. The following individuals involved in review of your submission have agreed to reveal their identity: Valerie Doye (Reviewer #1).

The reviewers have discussed the reviews with one another and the Reviewing Editor has drafted this decision to help you prepare a revised submission.

Summary:

This paper addresses the role of Crm1 in *HoxA* and *HoxB* activation in leukemic cells. The authors propose that chromatin-bound Crm1 functions as a platform to recruit a broad range of leukemic proteins. While the paper is potentially interesting to a broad audience, an important unanswered question is how CRM1 is recruited to *HOXA* and *B* loci. Does this depend on SET-Nup214 or NPM1c or not, and are these factors recruited in a cooperative fashion? This issue needs to be addressed (e.g., biochemically or with mutational studies) before the paper is suitable for publication in *eLife*.

Major comments:

1) What is the order of recruitment of Crm1 and SET-Nup214 or NPM1c?

Average CRM1 binding is very low in NPM1c peaks (Figure 5B and C), suggesting that NPM1c may often bind chromatin independently of CRM. CRM1 is extremely high in LOUCY HOX loci (Figure 1,2) were also SET-Nup214 is very high. Because SET is known to bind chromatin, SET-Nup214 may be the recruiter of CRM1 in this case. Importantly, the KPT-330 sensitivity does not discriminate between dependencies. The CRM1/NUP214 interaction in vitro is less sensitive to leptomycin B than the CRM1/NES interaction (e.g. Askjaer et al., 1999). However, the interactions within a chromatin complex may be more sensitive to overall CRM1 structure. Furthermore, the authors need to evaluate the effect of KPT330 on the total protein levels of CRM1 and NPM1 in the OC1-AML cells, before concluding on the impact of this drug on chromatin recruitment as changes in protein levels may explain the altered recruitment.

Additional experiments need to be added to address this important question. For example, the authors could specifically deplete SET-Nup214 from the LOUCY cells and assess its impact on CRM1 recruitment (by Chip-PCR) at a few specific loci.

2) Currently, the direct contribution of NPM1c on *HOX* gene activation has not been demonstrated and remains a correlation only.

The authors should determine the functional consequences of NPM1c (or NPM1 as control) overexpression on *Hox* gene expression and mESC differentiation similar to the experiments for Nup98-HoxA9 overexpression in the previous paper.

[Editors' note: further revisions were requested prior to acceptance, as described below.]

Thank you for submitting your article "Chromatin-bound CRM1 recruits SET-Nup214 and NPM1c onto *HOX* clusters causing aberrant *HOX* expression in leukemia cells" for consideration by *eLife*. Your article has been reviewed by two peer reviewers, and the evaluation has been overseen by a Reviewing Editor and James Manley as the Senior Editor. The following individuals involved in review of your submission have agreed to reveal their identity: Valerie Doye (Reviewer #1); Maarten Fornerod (Reviewer #2).

The reviewers have discussed the reviews with one another and the Reviewing Editor has drafted this decision to help you prepare a revised submission.

Summary:

The reviewers agree that in the revised version of the manuscript, the authors have performed several critical experiments that now answer to most of the points that were raised in the previous review. Thus, the paper is in principle suitable for publication; however, there are a number of points that the authors should address prior to acceptance.

Essential revisions:

1) Paragraph three in subsection “CRM1 is required for the recruitment of SET-Nup214 on *HOX* clusters and *HOX* gene activation”

To explain the milder sensitivity towards KPT-330 of MEGAL cells as compared to LOUCY cells for CRM1/SET-Nup214 binding and *Hox* gene activation, the authors indicate that "this is likely related to the fact that MEGAL cells show lower expression of SET-Nup214 than LOUCY" and refer for that to the paper from Port et al., 2016 in which unfortunately, the data were not shown (only the lower% of cells with foci was shown in that paper). Yet, the number of foci in the LOUCY and MEGAL cell lines may not be related to the expression level of the protein).

In contrast, comparing the two distinct blots provided in this study (Figure 3B for LOUCY and Figure 3—figure supplement 2A for MEGAL), it seems that the relative proteins levels of SET-Nup214 and NUP214 is 1:1 in LOUCY cells, but rather 2:1 or more in MEGAL cells, suggesting (assuming that the levels of endogenous Nup214 are comparable in both cell lines) that SET-Nup214 might be MORE abundant in the MEGAL cells.

Could the lower sensitivity of the MEGAL cells to KPT330 be due to an increased level of the fusion protein that would be less efficiently destabilized by the KPT330 treatment?

2) Paragraph five of subsection “CRM1 is required for the recruitment of SET-Nup214 on *HOX* clusters and *HOX* gene activation”: (and Figure 3—figure supplement 3) what about the impact of SET depletion on of *HOX* gene expression?

Why did the authors not complete the study of CRM1/Nup214 recruitment by the analysis of *HOX* gene expression? (Possibly because this was previously published? Or not required clearly enough in the previous review?). Was the result not expected?

3) In the same paragraph regarding the interpretation of the LMB experiment

"These results demonstrate that CRM1/Ran-GTP-/NES protein complex, which shows a higher affinity towards Nup214 and SET-Nup214 than CRM1 alone (ref to previous publications), is important for the recruitment of SET-Nup214 to *HOX* clusters, and its stabilization".

Here, this new experiment with LMB raises the question of which NES-containing protein(s) would be co-recruited along with CRM1 and SET-Nup214 on the *HOX* genes. Could the author clarify this paragraph? Also, the model in Figure 8 might take these aspects into account.

4) Regarding Figure 2A:

Was the western blot in Figure 2A (extracts in RIPA buffer) repeated in enough distinct experiments to ensure that the lower levels of Nup214 upon KPT-330 treatment in K562 cells are not an artifact of a specific experiment?

If not, best would be to remove this blot (that is not critical) and only show the total cell extract from Figure 2B. On the other hand, if Nup214 properties (extraction or stability) are specifically altered in K562 cells upon KPT330, this should be clearly stated in the text.

---

## [Author Response]

Major comments:1) What is the order of recruitment of Crm1 and SET-Nup214 or NPM1c?Average CRM1 binding is very low in NPM1c peaks (Figure 5B and C), suggesting that NPM1c may often bind chromatin independently of CRM. CRM1 is extremely high in LOUCY HOX loci (Figure 1,2) were also SET-Nup214 is very high. Because SET is known to bind chromatin, SET-Nup214 may be the recruiter of CRM1 in this case. Importantly, the KPT-330 sensitivity does not discriminate between dependencies. The CRM1/NUP214 interaction in vitro is less sensitive to leptomycin B than the CRM1/NES interaction (e.g. Askjaer et al., 1999). However, the interactions within a chromatin complex may be more sensitive to overall CRM1 structure. Furthermore, the authors need to evaluate the effect of KPT330 on the total protein levels of CRM1 and NPM1 in the OC1-AML cells, before concluding on the impact of this drug on chromatin recruitment as changes in protein levels may explain the altered recruitment.Additional experiments need to be added to address this important question. For example, the authors could specifically deplete SET-Nup214 from the LOUCY cells and assess its impact on CRM1 recruitment (by Chip-PCR) at a few specific loci.

We thank the reviewer for the suggestion. First, we performed western blotting using cell lysates from OCI-AML3 cells to evaluate the effect of KPT330 on the protein levels of CRM1 and NPM1c. As shown in Figure 5—figure supplement 1, our results showing that KPT330 caused a significant reduction of CRM1 protein, while the effect on NPM1c was subtler. Therefore, CRM1 most likely is a key molecule to recruit NPM1c to *HOX* cluster regions in OCI-AML3 cells. Our experiment using ES cells (Figure 7) also supports this hypothesis by showing that ectopically expressed NPM1c is targeted to *HOX* cluster regions, where the endogenous CRM1 already accumulates.

Next, we performed knockdown of SET-Nup214 to see whether it affects the binding of CRM1 to *HOX* clusters. Our ChIP-qPCR analysis demonstrated that the knockdown of SET-Nup214 did not significantly affect the CRM1 level bound at *HOX* cluster regions (Figure 3—figure supplement 3). This result suggests that CRM1 functions as a recruiter of SET-Nup214, as in the case of NPM1c.

In addition, we performed an experiment using leptomycin B (LMB), another well-known CRM1 inhibitor (Kudo et al., 1998). Unlike KPT330, LMB irreversibly binds to CRM1 (Sun et al., 2013) and does not induce the degradation of CRM1 protein (Turner et al., 2013)(Mendonca et al., 2014). Interestingly, we found that SET-Nup214 is also degraded by LMB treatment, as observed by KPT330 treatment (Figure 3—figure supplement 4). ChIP-qPCR revealed that LMB caused significant depletion of *HOX* cluster-bound CRM1 and SET-Nup214, and also exhibited down-regulation of *HOX* genes (Figure 3—figure supplement 4). These results demonstrate that the CRM1/Ran-GTP/NES protein complex, which shows a higher affinity toward Nup214 or SET-Nup214 than CRM1 alone (Askjaer et al., 1999) (Kehlenbach, Dickmanns, Kehlenbach, Guan, and Gerace, 1999) (Port et al., 2016), is important for the recruitment of SET-Nup214 to *HOX* clusters and its stabilization.

These results are mentioned in the text of the revised version as follows: “Next, to know the order of recruitment of SET-Nup214 and CRM1 onto *HOX* clusters, we performed knockdown of SET-Nup214. Reduction of SET-Nup214 did not significantly affect the amount of CRM1 that bound to *HOX* regions (Figure 3—figure supplement 3), suggesting that CRM1 functions as a recruiter of SET-Nup214.

In addition, we performed this experiment using LMB. Unlike KPT330, LMB irreversibly binds to CRM1 (Sun et al., 2013) and does not induce the degradation of CRM1 protein (Mendonca et al., 2014; Turner et al., 2013). Interestingly, we found that SET-Nup214 was also degraded by LMB treatment, as by KPT330 treatment (Figure 3—figure supplement 4). ChIP-qPCR revealed that LMB caused significant depletion of *HOX* cluster-bound CRM1 and SET-Nup214, and down-regulation of *HOX* genes (Figure 3—figure supplement 4). These results demonstrate that CRM1/Ran-GTP/NES protein complex, which shows a higher affinity toward Nup214 or SET-Nup214 than CRM1 alone (Askjaer et al., 1999; Kehlenbach et al., 1999; Port et al., 2016), is important for the recruitment of SET-Nup214 to *HOX* clusters, and its stabilization.”

2) Currently, the direct contribution of NPM1c on HOX gene activation has not been demonstrated and remains a correlation only.The authors should determine the functional consequences of NPM1c (or NPM1 as control) overexpression on Hox gene expression and mESC differentiation similar to the experiments for Nup98-HoxA9 overexpression in the previous paper.

Our qPCR analysis revealed that *Hox* gene is not upregulated in NPM1c expressing ES cells (see Author response image 1), even though NPM1c accumulates at four *Hox* cluster loci (Figure 7B). We speculate that the *Hox* cluster region in ES cells may possess a different chromatin structure or epigenetic status as compared with that of OCI-AML3 cells and may be difficult to be activated solely by the ectopic expression of NPM1c.

As mentioned in the manuscript, a previous paper (Brunetti et al., 2018) has clearly demonstrated that NPM1c is involved in *HOX* gene activation in OCI-AML3 leukemia cells. Especially, their results showed that the targeted degradation of NPM1c by using FKBP-based degron system resulted in an immediate down-regulation of *HOX* genes, strongly suggesting the direct contribution of NPM1c on *HOX* gene activation. Further, they showed that both the disruption of NES in NPM1c and treatment with KPT330, which cause dissociation of NPM1c from *HOX* cluster chromatin region as revealed by our study, significantly suppressed *HOX* genes. Together, our paper strongly suggests that NPM1c directly binds to *HOX* cluster to activate *HOX* genes in OCI-AML3 cells.

To further provide evidence for the relation between NPM1c/CRM1 and *HOX* gene activation, we monitored the effect of KPT330 on active Pol II on *HOX* cluster region in OCI-AML3. Our results revealed that both the S2P and S5P forms of active Pol II showed a significant sensitivity toward KPT330 treatment, as observed in SET-Nup214 expressing LOUCY cells (added in this revised manuscript as Figure 5—figure supplement 2). The robust signal of active Pol II, among the regions we examined, was observed at *HOXA9* region, which correlated well with CRM1 and NPM1c binding (Figure 5D and 5E). Furthermore, we found that previous RNA-seq data and ChIP-seq data of active histone marks, such as H3K27Ac or H3K4me3 in OCI-AML3 cells (Brunetti et al., 2018), are well-correlated with the binding pattern of CRM1/NPM1c (Figure 5—figure supplement 3). Especially, the binding signals of NPM1c/CRM1 and active histone marks were high in posterior (AbdB-related) *HOX* genes, at *HOXA* cluster (*HOXA9* to *HOXA13*). Collectively, NPM1c is most likely involved in the activation of Pol II together with CRM1 in OCI-AML3.

Considering the above-mentioned issues, we would like to change the title of the paper to “Chromatin-bound CRM1 recruits SET-Nup214 and NPM1c onto *HOX* clusters causing aberrant *HOX* expression in leukemia cells”.

These results are mentioned in the text of the revised version as follows: “However, gene expression analysis revealed that NPM1c could not activate *Hox* gene in ES cells (data not shown). We speculate that *Hox* cluster region in ES cells may possess a different chromatin structure or epigenetic status as compared with that of OCI-AML3 cells and may be difficult to be activated solely by the ectopic expression of NPM1c. Further study will be needed to determine the exact mechanism of gene activation by NPM1c.”; and “In addition, both S2P and S5P forms of active Pol II showed significant sensitivity toward KPT330 treatment, as observed in SET-Nup214 expressing LOUCY cells (Figure 5—figure supplement 2). The robust signal of active Pol II was observed at *HOXA9* region, which correlated well with CRM1 and NPM1c binding (Figure 5D). Furthermore, previous RNA-seq data and ChIP-seq data of active histone marks in OCI-AML3 cells (Brunetti et al., 2018) correlated well with the binding pattern of CRM1/NPM1c (Figure 5—figure supplement 3). Especially, the binding signals of NPM1c/CRM1 and active histone marks were enriched in posterior (AbdB-related) *HOX* genes in the *HOXA* cluster (*HOXA9* to *HOXA13*).”

[Editors' note: further revisions were requested prior to acceptance, as described below.]

Essential revisions:1) Paragraph three in subsection “CRM1 is required for the recruitment of SET-Nup214 on HOX clusters and HOX gene activation”To explain the milder sensitivity towards KPT-330 of MEGAL cells as compared to LOUCY cells for CRM1/SET-Nup214 binding and Hox gene activation, the authors indicate that "this is likely related to the fact that MEGAL cells show lower expression of SET-Nup214 than LOUCY" and refer for that to the paper from Port et al., 2016 in which unfortunately, the data were not shown (only the lower% of cells with foci was shown in that paper). Yet, the number of foci in the LOUCY and MEGAL cell lines may not be related to the expression level of the protein).In contrast, comparing the two distinct blots provided in this study (Figure 3B for LOUCY and Figure 3—figure supplement 2A for MEGAL), it seems that the relative proteins levels of SET-Nup214 and NUP214 is 1:1 in LOUCY cells, but rather 2:1 or more in MEGAL cells, suggesting (assuming that the levels of endogenous Nup214 are comparable in both cell lines) that SET-Nup214 might be MORE abundant in the MEGAL cells.Could the lower sensitivity of the MEGAL cells to KPT330 be due to an increased level of the fusion protein that would be less efficiently destabilized by the KPT330 treatment?

We thank the reviewer’s for their comments. First of all, in Figure 3B (new Figure 3A) and Figure 3—figure supplement 2A, we used different methods to prepare cell lysates, thus, these figures are not suitable for a comparison. As stated in the figure legends, western blotting in Figure 3B (new Figure 3A) was performed using cell lysates prepared by boiling the cells in a sample buffer. However, in Figure 3—figure supplement 2A (MEGAL), we used RIPA buffer to prepare cell lysates. We are still unsure why, but cell lysates prepared in RIPA buffer have a tendency to show a higher ratio of SET-Nup214/Nup214 as compared with the ratio obtained when the samples were prepared by boiling cells in the sample buffer.

Western blotting of LOUCY cell lysate prepared with RIPA buffer is shown in Figure 3A (new Figure 3—figure supplement 5A). This clearly showed that the ratio of SET-Nup214/Nup214 in LOUCY cells was higher than that observed in MEGAL cells in Figure 3—figure supplement 2A. These results demonstrate that MEGAL cells express lower amounts of SET-Nup214 than LOUCY cells. However, we still have not figured out why the SET-Nup214 in MEGAL cells shows lower sensitivity to KPT330 as compared with the sensitivity in LOUCY cells.

To further clarify the differences in SET-Nup214 expression levels, we compared the cell lysates obtained from LOUCY and MEGAL cells, which were prepared either by RIPA buffer or boiling in a sample buffer, in the same blot (Author response image 2).

**Author response image 2. respfig2:** 

We changed the text as follows:

“The difference between LOUCY and MEGAL cells in the degree and region of CRM1/SET-Nup214 binding and gene activation is likely related to the fact that MEGAL cells show lower expression of SET-Nup214 than LOUCY cells [comparing the ratio of SET-Nup214/Nup214 in Figure 3A—figure supplement 5A (LOUCY) and Figure 3—figure supplement 2A (MEGAL)]. However, the reason for the lower sensitivity of SET-Nup214 in MEGAL cells to KPT330 remains unknown.”

2) Paragraph five of subsection “CRM1 is required for the recruitment of SET-Nup214 on HOX clusters and HOX gene activation”: (and Figure 3—figure supplement 3) what about the impact of SET depletion on of HOX gene expression?Why did the authors not complete the study of CRM1/Nup214 recruitment by the analysis of HOX gene expression? (Possibly because this was previously published? Or not required clearly enough in the previous review?). Was the result not expected?

We have not included the data since it was previously published. As shown in Figure 3—figure supplement 3, we observed that the knockdown of SET-Nup214 caused down-regulation of *HOX* genes in LOUCY cells, in agreement with a previous paper (Van Vlierberghe et al., 2008).

We changed the text as follows:

“Reduction of SET-Nup214 caused down-regulation of *HOX* genes (Figure 3—figure supplement 3) in agreement with a previous report (Van Vlierberghe et al., 2008), but did not significantly affect the amount of CRM1 that bound to *HOX* regions, suggesting that CRM1 functions as a recruiter of SET-Nup214 (Figure 3—figure supplement 3).”

3) In the same paragraph regarding the interpretation of the LMB experiment"These results demonstrate that CRM1/Ran-GTP-/NES protein complex, which shows a higher affinity towards Nup214 and SET-Nup214 than CRM1 alone (ref to previous publications), is important for the recruitment of SET-Nup214 to HOX clusters, and its stabilization".Here, this new experiment with LMB raises the question of which NES-containing protein(s) would be co-recruited along with CRM1 and SET-Nup214 on the HOX genes. Could the author clarify this paragraph? Also, the model in Figure 8 might take these aspects into account.

Thank you for the comments. We added the sentence as follows: “However, it is currently unknown which NES-containing protein(s) would be co-recruited with CRM1 and SET-Nup214 on the *HOX* genes.”

We also modified the original Figure 8, and added a sentence to the figure legend as follows: “In the case of Nup98HoxA9 and SET-Nup214, the NES-containing protein(s) is/are most likely involved in the complex.”

4) Regarding Figure 2A:Was the western blot in Figure 2A (extracts in RIPA buffer) repeated in enough distinct experiments to ensure that the lower levels of Nup214 upon KPT-330 treatment in K562 cells are not an artifact of a specific experiment?If not, best would be to remove this blot (that is not critical) and only show the total cell extract from Figure 2B. On the other hand, if Nup214 properties (extraction or stability) are specifically altered in K562 cells upon KPT330, this should be clearly stated in the text.

Thank you for the suggestion. We carefully repeated the western blotting, but we could not confirm the effect of KPT330 on Nup214 in K562 cells. We deeply apologize for the confusion. As suggested, we have removed that particular blot from the manuscript (original Figure 2A).